



# Simulating carbon fluxes in boreal catchments: WSFS-Vemala model development and key insights

Marie Korppoo[1], Inese Huttunen[1], Markus Huttunen[1], Maiju Narikka[1], Jari Silander[2], Tom Jilbert[3], Martin Forsius[4], Pirkko Kortelainen[4], Niina Kotamäki[5], Cintia Uvo[5,6], Anna-Kaisa Ronkanen[5]

[1] Climate solutions, Finnish Environment Institute (Syke), Helsinki, Finland
[2] Quality of information, Finnish Environment Institute (Syke), Helsinki, Finland
[3] Environmental geochemistry group, Department of geosciences and geography, University of Helsinki, Finland
[4] Nature solutions, Finnish Environment Institute (Syke), Helsinki, Finland
[5] Marine and Freshwater solutions, Finnish Environment Institute (Syke), Helsinki, Finland
[6] Water Resources Engineering, Lund University, Sweden

*Correspondence to*: Marie Korppoo (marie.korppoo@syke.fi)

**Abstract.**

Lakes and streams play an important role in the global carbon cycle through carbon sedimentation and evasion. The
development of carbon processes in the water quality model WSFS-Vemala (Vemala) presents a significant advancement in simulating carbon dynamics, particularly in capturing both total organic (TOC) and inorganic (TIC) carbon processes and their contributions to carbon retention and emissions through a river/lake network. The model was tested in the Vantaanjoki catchment, located in southern Finland and covering an area of 1680 km$^2$. The model's ability to simulate TOC and TIC loading across various land use and soil types aligns closely with reported literature values. The addition of organic acids to the total
alkalinity definition improved pH simulations and thus the simulation of $CO_2$ emissions in the acidic and organic rich waters of Finland. Annual $CO_2$ emissions of 25 gC m$^2$ yr$^{-1}$ were simulated from lake Tuusulanjärvi, the largest lake in the catchment, and 223-260 gC m$^{-2}$ yr$^{-1}$ from the river network, while only 3 gC m$^{-2}$ yr$^{-1}$ was simulated as organic carbon burial in the lake sediments. The model's performance in estimating $CO_2$ emissions shows good correlations with established ranges for lakes as well as good correlation with TOC and TIC loads across the river network. The inclusion of sedimentation and
mineralization processes in the lake carbon budget underlines the necessity of accounting for both organic and inorganic pathways in carbon modelling. This improved representation of the carbon cycling in Vemala, linked with the phytoplankton growth and nutrient cycling, allow to distinguish between carbon losses to the atmosphere and long-term carbon storage in the sediments of inland waters. Overall, the enhanced Vemala model provides a robust foundation for understanding carbon cycling and supporting sustainable, integrated water resource management and scenario assessments from sub-catchments to
the national scale.



## 1 Introduction

Addressing the global challenge of climate change requires both mitigation and adaptation strategies, supported by high-quality observational data and advanced computational models. A critical aspect of this effort is understanding the processes that govern carbon (C) dynamics within catchments, including accumulation, sequestration, release, and transport. The intricate
interplay between eutrophication and carbon accumulation/release in aquatic ecosystems remains a subject of limited understanding, posing challenges to effective river basin management (Ward et al., 2017). The importance of C budget estimates for comprehensive river basin management plans cannot be overstated, as it plays a crucial role in evaluating the effectiveness of carbon-related interventions. Lakes and rivers play a pivotal role in transporting, capturing and releasing dissolved organic carbon (DOC) and dissolved inorganic carbon (DIC) from terrestrial sources (Cole et al., 2007; Rantakari,
2010; Mendonça et al., 2017; Tranvik et al. 2018; Amora-Nogueira et al. 2022; Algesten et al., 2003; Allesson et al., 2022). At the global scale, catchment flux from land to sea of DOC is equivalent to DIC (Hope et al., 1994) and freshwater systems are supersaturated in $CO_2$ (Cole et al., 1994). However, in boreal environments DOC fluxes are greater than DIC. Specifically for the Baltic Sea, DOC represents up to 80% of the carbon export to the Sea (Räike et al., 2016). A clear correlation exists between DOC concentration in lake water and $CO_2$ supersaturation (Sobek et al., 2005), especially in lakes surrounded by
forests (Rantakari and Kortelainen, 2008), suggesting that humic lakes, which are prevalent in boreal environments (Kortelainen, 1993), may significantly influence carbon gas emissions in these regions (Rantakari and Kortelainen, 2008, Kortelainen et al. 2006, Forsius et al. 2017). Holmberg et al. (2023) estimated, using area-specific empirical emission coefficients, that greenhouse gas (GHG) emissions from lakes and rivers in Finland currently are about 13.4 $TgCO_2$eq $yr^{-1}$ or about 9% of total Finnish GHG emissions to the atmosphere. DIC in freshwater is composed of three main species: bicarbonate
($HCO_3^-$), carbonate ($CO_3^{2-}$) and dissolved carbon dioxide ($H_2CO_3$ or $CO_2$aq.) and is primarily derived from rock weathering and mineralisation of organic carbon. Due to the equilibrium reactions in the carbonate system, speciation of DIC is tightly linked to pH. Therefore, pH impacts on dissolved $CO_2$ concentration and thus $CO_2$ emissions through the water-atmosphere interface (Jarvie et al., 2017). Boreal catchments in Finland are characterised by non-carbonate bedrocks leading to low buffer capacity of the freshwater (low alkalinity and pH) that enhances the availability of dissolved $CO_2$ for evasion (Tranvik et al.,
55  2009).

Recent developments in mathematical modeling of river basin biogeochemistry include the integration of DOC dynamics (e.g. INCA-C (Futter et al., 2007), SWAT-C (Qi et al., 2020)). However, DIC dynamics have been omitted from most models, despite the potential value of adding such processes to quantify carbon emissions from aquatic ecosystems (e.g. Jarvie et al., 2017, Marescaux et al., 2020). The THINCARB model (Jarvie et al., 2017) calculates DIC ($HCO_3^-$, $CO_3^{2-}$ and $H_2CO_3$) and the
excess partial pressure of carbon dioxide in a water sample based on pH and alkalinity measurements available in the Harmonised Monitoring Scheme of Great Britain. Marescaux et al. (2020) added an inorganic carbon module to the process-based biogeochemical model pyNuts-Riverstrahler model to quantify $CO_2$ emissions from the whole hydrographic network of the Seine catchment. In both models, alkalinity is calculated based primarily on the carbonate system. However, in low





carbonate alkalinity regions with high DOC concentrations, organic acid anions can increase alkalinity (Hruska et al., 2003).
Moreover, in acidic and organic rich waters such as found in boreal regions, the buffering capacities of the carbonate system are lower, which increases the sensitivity of calculated $CO_2$ concentrations and emissions (Abril et al., 2015).

The WSFS-Vemala model (Vemala; Huttunen et al. 2016), is a water quality and nutrient load model system developed by the Finnish Environment Institute (Syke) for Finnish watersheds. Vemala is operative and widely utilized in Finland in planning the implementation of the Water Framework Directive (WFD; 2000/60/EC) by the regional ELY Centres (Centres for
Economic Development, Transport and the Environment in Finland) for both inland waters (Huttunen et al. 2016) and coastal waters (Lignell et al., 2024). While nutrient cycling is well incorporated into this model (Korppoo et al., 2017), regarding both terrestrial and aquatic processes, the same level of detail is lacking for C. This omission leaves a significant gap in our ability to support environmental permitting decisions, manage carbon related activities, and deepen our understanding of carbon dynamics. A pivotal aspect that demands exploration is the connection between nutrients and carbon, especially in the context
of reducing nutrient loading in aquatic environments. If we reduce nutrient pollution without understanding its effect on carbon, we might unintentionally increase carbon emissions or miss opportunities to store carbon in aquatic systems. This connection is key for designing better environmental policies and carbon management strategies.

In this paper, we quantify carbon losses through the river and lake network by modelling total carbon (TC) processes including both total organic carbon (TOC) and total inorganic carbon (TIC) in the terrestrial and aquatic environments of Lake
Tuusulanjärvi (5.9 km$^2$) and the entire Vantaanjoki catchment (1680 km$^2$), a Finnish river basin flowing to the Baltic Sea. For this purpose, we developed the TOC terrestrial model, based on Vemala-N model (Huttunen et al., 2016) and INCA-C model (Futter et al., 2007). The model represents three carbon storages in the soil linked to soil types and land uses (Fig.1), and uses annual litter fall (Minunno et al., 2019) and initial C storage in soil to allow for the simulation of DOC load transported from the soil and into the river network. Moreover, we developed the leaching of TIC using alkalinity as a proxy based on soil types
to simulate the effect of rock weathering on TIC leaching. Finally, we added TIC as a variable in the existing Vemala biogeochemical submodel (Korppoo et al., 2017) and the processes affecting TIC in the water column including phytoplankton growth, TOC mineralization from water and sediments, and pH and alkalinity effects on the $CO_2$ emissions across the water-air interface. The total alkalinity in the model was defined as carbonate and organic alkalinity to improve the representation of pH and therefore the simulation of $CO_2$ concentrations in the water column in organic rich waters.


This improved representation of the carbon cycling in Vemala, linked with the phytoplankton growth and nutrient cycling will allow us to specify the carbon losses in the catchment including the distinction between carbon losses to the atmosphere and long-term carbon storage in the sediments. Our aim is to quantitatively predict the importance of carbon losses at the scale of the whole aquatic ecosystem. For this purpose, we first assess the model's performance in the Vantaanjoki catchment in
southern Finland by comparing the results to the observed daily total organic and inorganic carbon loads and to literature values. In a second step, we focus on the simulation of carbon sequestration in and emissions from the largest lake in the basin (Tuusulanjärvi) and the river network at a daily/annual time step. Finally, we run a sensitivity analysis of the model to examine





how variations in mineralisation rates, phytoplankton growth and $CO_2$ air-water transfer rates affect carbon losses from the aquatic ecosystem. This analysis will help to identify the most influential processes and improve our understanding of carbon

dynamics in the system.

## 2 Model description

### 2.1 Vemala hydrological submodel

A physically based hydrological model (WSFS: Watershed Simulation and Forecasting System; Jakkila et al., 2014), is integrated into Vemala (Fig. 1). The unsaturated soil layer is represented by a 2-layer soil moisture model (surface and sub-

surface), and a soil saturated layer characterizing groundwater. Water infiltrates from the surface layer to the sub-surface layer if the soil moisture content exceeds field capacity. The sub-surface layer produces the main part of the runoff to rivers and lakes and of the percolation to the groundwater (Jakkila et al., 2014). Runoff is produced from the surface layer only when the ground is frozen.

This integration enables the simulation of hydrological processes that vary on mineral and organic soils, accounting for distinct

hydraulic conductivities and water holding properties of mineral and peat soils. Finnish soil maps (Finnish Soil Database, Natural Resources Institute Finland & Geological Survey of Finland, 2023, Fig. 3c) are used for soil type distribution. Specifically, six soil types - organic, clay, till, silt, sand, and rocky soils - have been used as input to the model for each small brook catchment. Small brook catchments are defined by a river width greater than 2m (Metadata: Shoreline10 from Syke and National Land Survey of Finland, MML). The median size of the catchments in Vemala is 0.8km² with 80% of the catchments

smaller than 2km². Corine Land Cover data (Fig. 3b) was used to classify each small brook catchment into land use classes - field, forested, bog, or built areas. This classification is essential for simulating evapotranspiration accurately.

There are considerable differences in hydrological conditions and organic carbon content in mineral and peat soils, leading to different TOC leaching patterns. These differences have been considered both in the hydrological and the terrestrial TOC process-based models (Fig. 1). In peat soils, shallow groundwater can contribute the most to runoff through baseflow,

especially during prolonged dry periods (Mozafari et al., 2023). Shallow water tables with small range of fluctuation are characteristic for peat soils. Shallow groundwater can have significantly higher DOC concentrations than deep groundwater. This pattern probably results from occurrence and movement of shallow groundwater through the organic-rich surficial deposits (Gibson et al., 2000). By definition, peat soils contain more organic matter (> 40%, Lemola et al., 2018) than mineral soils, and SOC (solid organic C) vertical distribution is different in mineral and peat soils. In mineral soils the SOC content

generally decreases with depth, whereas in organic soils it increases with depth (Hiederer, 2009).





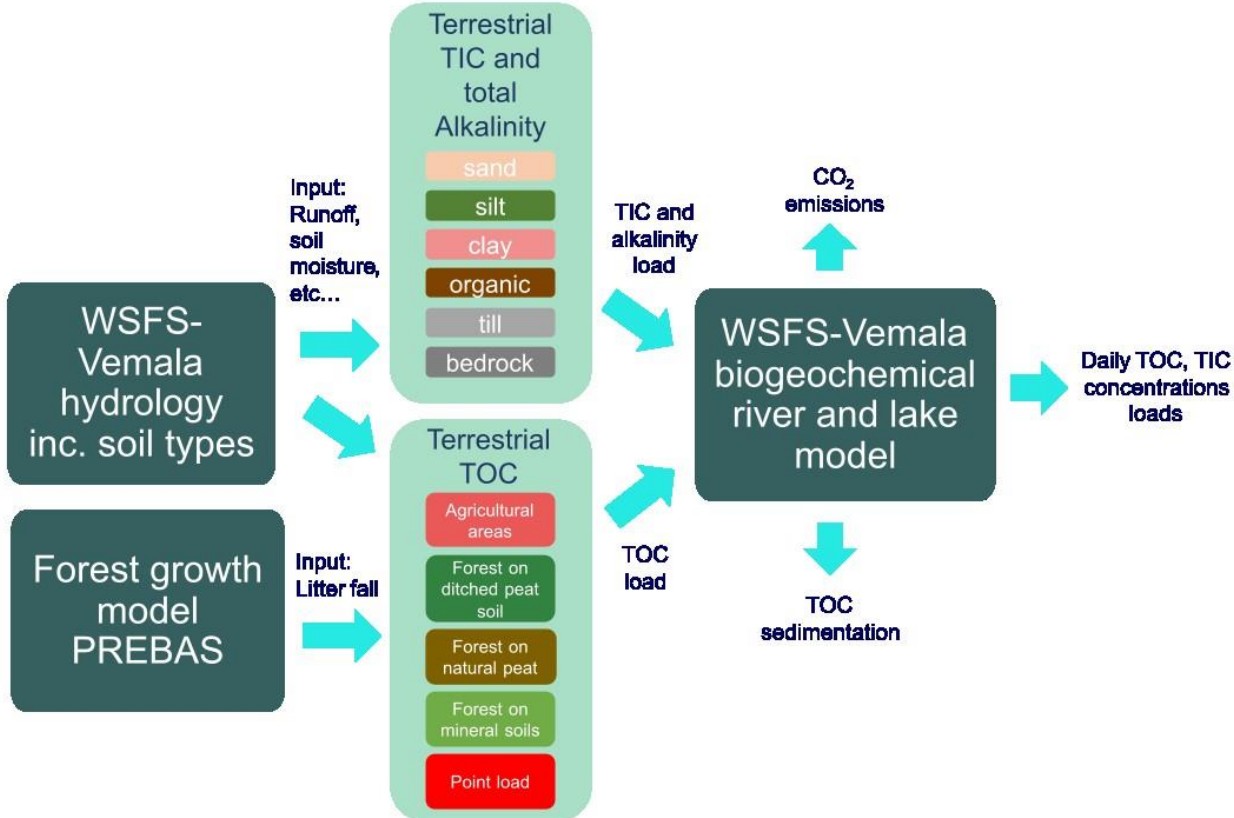

**Figure 1: Schematic representation of the structure of the Vemala carbon model**

## 2.2 Terrestrial carbon submodels

Vemala TOC terrestrial model development (Fig. 1) is based on Vemala-N model (Huttunen et al., 2016) and INCA-C model (Futter et al., 2007). There are three C storages in the soil – SOC, DOC and DIC linked to soil types and land uses. Inputs to the model are annual litter fall and initial C storage in soil. Interactions among these pools are as follows:

- SOC can be disassociated into DOC, and vice versa.

- SOC and DOC can be mineralized into DIC that is simulated as a loss from the system to the air.

- DOC leaches with the subsurface runoff and baseflow.

There are six land use classes in the TOC model – agriculture on clay, coarse and peat soils, forest on mineral soils and ditched and natural peat soils. C mass balance of SOC and DOC storages for two soil layers (unsaturated soil layer and groundwater layer) is simulated daily. Leaching of DOC ($kg^{-1}$ $ha^{-1}$ $d^{-1}$) is calculated as follows: mass of DOC in layers (kg $ha^{-1}$) is divided by soil moisture or groundwater storage (mm) to obtain the simulated concentration. DOC concentration is then multiplied by

daily runoff (mm $d^{-1}$) from each soil layer to obtain the DOC load (kg $ha^{-1}$ $d^{-1}$) transported out from each layer (Eq. (1) and Eq.



(2)). $cf_{moisture}$, $cf_{gw}$ are plant unavailable water and groundwater adjustment parameters, which are manually calibrated for each land use/soil texture class.

$$DOC_{leaching,1} = \frac{m_{DOC,1}}{soil_{moisture}+cf_{moisture}} * runoff_1 \qquad (1)$$

$$DOC_{leaching,2} = \frac{m_{DOC,2}}{groundwater_{storage}+cf_{gw}} * gw_{flow_2} \qquad (2)$$

At the national scale, the spatial variability in the input data has been considered in the TOC model. Initial C content in agricultural soils is based on field parcel data from soil laboratory Viljavuuspalvelu oy which contains soil organic matter (SOM) class (vm - low, m – medium, rm –rich, erm – very rich, mm – mull, Tm – peat soil). Initial C content in mineral forest soils is based on Finnish multisource national forest inventory data (Mäkisara et al., 2016). Litter fall data for forests is obtained from PREBAS model results for 16x16m grids for all Finland. PREBAS combines carbon acquisition through photosynthesis,

tree growth and soil carbon processes (Minunno et al., 2019). DOC production rates for peat soils are based on Laurén et al. (2019), and temperature dependency coefficient Q10 values, that describes relative change of C release rate under a 10-degree change in temperature, are calibrated.

Vemala TIC terrestrial model development (Fig. 1) is based on a regression model between TIC concentration, using alkalinity as a proxy, and runoff per soil type. TIC is assumed fully dissolved and thus is representing DIC. The alkalinity per soil type

($Alk_n$) is calibrated with runoff from each soil type ($qr_n$) in the catchment area (Eq. (3)):

$$Alk_n = \gamma_n * qr_n^{\varepsilon_n} \qquad (3)$$

with $\gamma_n$ and $\varepsilon_n$ the alkalinity coefficients for each soil type. Negative alkalinities are excluded from this model. The range of alkalinities per soil types was defined from Korkka-Niemi (2001) measurements of well waters in Finland using the mean values per soil type with a range of ±20% (Table 2).

**2.3 Aquatic biogeochemical submodel**

The Vemala biogeochemical submodel (Korppoo et al., 2017) was developed to couple organic carbon to inorganic carbon processes and simulate total carbon cycling. The aim is to simulate organic carbon long-term sedimentation or sequestration and $CO_2$ emissions to the air separately within the aquatic ecosystem (Fig. 2). The new state variables are TIC, alkalinity and pH. TIC represents the sum of three fractions ($CO_2$, $HCO_3^-$, $CO_3^{2-}$), TOC the sum of four fractions ($DOCH_3$, $DOCH_2^-$, $DOCH^{2-}$

and $DOC^{3-}$), and pH the hydrogen ions $H^+$ (Eq. (4)):

$$pH = -\log[H^+] \qquad (4)$$

The processes affecting C and simulated in Vemala are mineralisation that transforms TOC into TIC, photosynthesis that consumes TIC for algal growth, lysing, grazing and viruses that release TOC to the water, algal sedimentation as TOC, TOC compaction in the sediments representing long-term sedimentation, and exchange of $CO_2$ through the water-air interface (Fig.

2). In Finnish waters, most of the TOC is dissolved (Kortelainen et al., 2006, Mattsson et al., 2005) and thus direct sedimentation of TOC is neglected in the model and TOC is considered equivalent to DOC. However, TOC sedimentation happens through phytoplankton settling, thus feeding the sediments with organic matter.



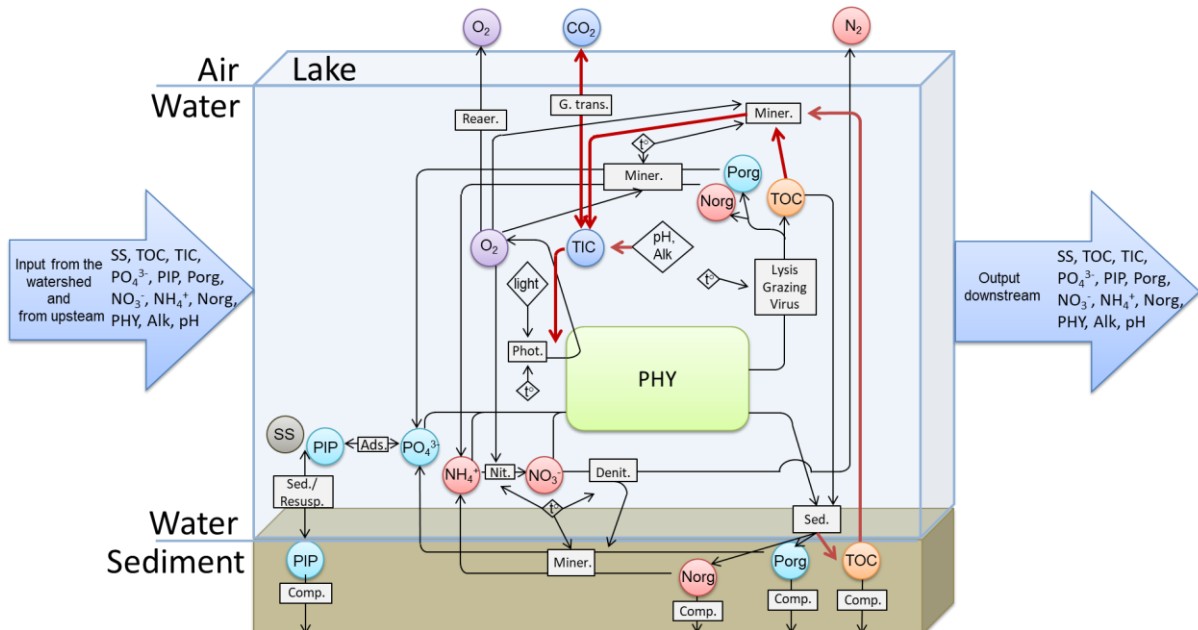

SS: Suspended sediments, TOC: Total organic carbon, TIC:Total inorganic carbon, $O_2$: Oxygen, $CO_2$: Carbon dioxide, $PO_4^{3-}$: Phosphate, PIP: Particulate inorganic phosphorus, Porg: Organic phosphorus, $NO_3^-$: Nitrate, $NH_4^+$: Ammonium, Norg: Organic nitrogen, $N_2$: Nitrogen gas, PHY: Phytoplankton, Alk: Alkalinity
Reaer.: Reaeration, Miner.: Mineralisation, G.trans.: Gas transfer, Sed.: Sedimentation, Resus.: Resuspension, Ads.: Adsorption, Nit.: Nitrification, Denit.: Denitrification, Comp. Compaction, Phot.: Photosynthesis

**Figure 2: Vemala v3 biogeochemical model including the inorganic carbon cycling with new processes shown on the figure in red. Adapted from Korppoo et al. (2017).**

The availability of TIC for release to the atmosphere depends on pH, alkalinity and TOC. pH models consider the acidity of the main acids in the water. The simple structure of pH models in freshwater use carbonate, oxide ions and protons to define the alkalinity (e.g. Munhoven, 2020, Marescaux et al., 2020). However, this type of model proved too simplistic for Finnish waters with high DOC and low concentrations of inorganic carbon (Abril et al. 2015). Hruska et al. (2003) quantified the acidity of dissolved organic carbon using three dissociation constants ($pKa_1$=3.04, $pKa_2$=4.51, $pKa_3$=6.46, Fig. A1) and a site density, which represents the number of carboxylic groups per milligram of DOC, of 10.2 microequiv $mg^{-1}$ of DOC +/- 0.6 for Swedish experimental sites. The pH buffering by DOC is thus significant in the pH range of 3.5-6.5. DOC can thus be simplified to a triprotic model with four fractions. $DOCH_3$ is omitted from this model as it is outside the range of pH for natural freshwaters. The resultant alkalinity definition in this model can be expressed as (Eq. (5)):

$$Alk = (\alpha_1 + 2\alpha_2)[DIC] + [OH^-] - [H^+] + (\beta_1 + 2\beta_2 + 3\beta_3)[DOC] \quad (5)$$

With $\alpha_1$ and $\alpha_2$ the respective proportion of $HCO_3^-$ and $CO_3^{2-}$ ions in DIC and $\beta1$, $\beta2$ and $\beta3$ the respective proportion of $DOCH_2^-$, $DOCH^{2-}$ and $DOC^{3-}$ ions in DOC. $OH^-$ and $H^+$ are the concentrations of hydroxide ions and protons and Alk refers to alkalinity. The proportion of $CO_2$ compared to TIC in the water depends on pH, alkalinity and TOC. This equilibrium between



TOC, TIC, Alk and pH is solved using the Newton-Raphson algorithm after reformulating Eq. (5) to solve for [H$^+$] using a polynomial equation as in Munhoven 2013 (Appendix A: pH model). Once pH is determined, $CO_2$ concentration in the water is calculated. The $CO_2$ gas exchange rate (F$_{CO2}$, in mgC m$^{-2}$ d$^{-1}$) at the water-atmosphere interface is calculated using (Eq. (6)):

$$F_{CO2} = k_{CO2} \times (CO2 - CO_{2atm}) \qquad (6)$$

with the concentration of $CO_2$ in the water column ($CO_2$, mgC m$^{-3}$) and at equilibrium with the atmospheric $CO_2$ ($CO_{2atm}$, mgC m$^{-3}$) (Eq. (12)) and k$_{CO2}$ the gas transfer rate at the water atmosphere interface (m d$^{-1}$, Eq. (7)):

$$k_{CO2} = k_{600} \times \left(\frac{Schmidt}{600}\right)^{-0.6667} \times (1 - z_{ice}) \qquad (7)$$

with k$_{600}$ the transfer velocity (Eq. (9) for the transfer velocity in a lake or Eq. (10) for the transfer velocity in a river) for the Schmidt number (cm h$^{-1}$), Schmidt is the Schmidt number for $CO_2$ (Eq. (8)) and z$_{ice}$ the fraction of the lake covered by ice.

This equation assumes that ice works as a cap on lakes impeding the exchange of gas at the water-atmosphere interface.

$$Schmidt = 1911.1 - 118.11 \times T_c + 3.4527 \times T_c{}^2 - 0.041320 \times T_c{}^3 (8)$$

with T$_c$ the temperature in Celsius (Wanninkhof, 1992).

$$k_{600lake} = \max(0.0, -1.318 + 2.067 \times xwind) \qquad (9)$$

k$_{600lkake}$ is the transfer velocity for the Schmidt number 600 (cm h$^{-1}$) for $CO_2$ and xwind the wind speed at 10 m height (m s$^{-1}$)

from Jonsson et al. (2008) for lakes.

$$k_{600river} = 13.82 + 0.35 * vel \qquad (10)$$

with vel the water velocity (cm s$^{-1}$) and k$_{600river}$ the transfer velocity for the Schmidt number 600 (cm h$^{-1}$) for rivers up to a width of 100m (Eq. (10), Alin et al., 2011).

The atmospheric concentration of $CO_2$ has steadily increased in the past decades (Fig. A2). Data from the ICOS website

(Integrated Carbon Observation System ICOS RI, licensed under CC4BY) allowed the modelling of the daily $CO_2$ concentration in the atmosphere (Eq. (11), r$^2$=0.89).

$$pCO_2 = 362 + 2.3 \times (Year - 1997) + 15 \times \sin\left(\frac{Date+60}{365.2425 \times 2 \times \pi}\right) \qquad (11)$$

with pCO2, the atmospheric $CO_2$ (in ppm) as simulated based on the Hyytiälä and Pallas stations in Finland, Year is the simulation year and Date is the day of the year (1-365).

$$CO_{2atm} = pCO_2 \times K_0 \times M_C \times p_{atm} \times dens \qquad (12)$$

In which $CO_{2atm}$ is the $CO_2$ concentration at saturation in the water (mgC m$^{-3}$) and K$_o$ the solubility of $CO_2$ in the water (molC kg$^{-1}$ atm$^{-1}$), M$_c$ the molar mass of carbon (12 gC mol$^{-1}$) and p$_{atm}$ the atmospheric pressure and pCO$_2$ the atmospheric concentration of $CO_2$ (Eq. (11)), and dens the density of the water as a function of the temperature (Chen and Millero, 1976).

## 2.4. Calibration and validation

The Vemala calibration process used a modification of the direct search Hooke-Jeeves optimisation algorithm as described in Huttunen et al. (2016). The model was calibrated over the period January 2004 to December 2023, while the period January



1990 to December 2003 was used for the validation of the model. The Nash-Sutcliffe criterion (NSE) was used to evaluate the model performance; the NSE values were calculated separately for the calibration and validation. The NSE coefficient ranges from $-\infty$ to 1. A coefficient of 1 corresponds to a perfect match between simulated and observed loads, while a coefficient of 0 presents a model that is as accurate as the mean of the observed data. Vemala calibration is a process starting with the calibration of first the hydrological submodel followed by the terrestrial submodels and ending with the biogeochemical submodel. Vemala TOC model is calibrated in two steps: first, there is a manual calibration of parameters for each of the six land use classes against annual TOC exports reported in literature. Table A1 shows the annual exports of TOC used in guiding the manual calibration. However, it is very hard to find national scale TOC export values because TOC export highly variates depending on the soil types, vegetation and along a South-North gradient in Finland. Export equations by Finér et al. (2021) are used for forests on mineral soils and peatlands, however these equations only consider the South-North gradient. The second step of the calibration is an automatic calibration of two parameters – coefficients for adjusting DOC production rates for upper and lower soil layer. Vemala TIC terrestrial model is calibrated via the alkalinity proxy. Alkalinity calibration is done automatically with two parameters for each soil type (cf and exp in Eq. (3)) at the Vantaanjoki catchment scale using all the alkalinity observations over the period 2004-2023. Finally, water quality observation points monitored regularly in the catchment are used in the calibration of the biogeochemical model (Fig. 5, Korppoo et al., 2017).

### 2.5 Sensitivity analysis

The biogeochemical deterministic model developed in this study is built on well-known enzyme-catalysed reactions (e.g. Michaelis-Menten) that focus on simulating the limiting reactions of the system affected by environmental conditions (e.g. light, temperature, flow). The range of parameters used in this application have been described in Korppoo et al. (2017) and are based on experimental measures or range from the literature. In this study, we run a sensitivity analysis of the latest processes added to the model in this development and affecting carbon cycling: mineralisation that transforms TOC into TIC, phytoplankton growth that through photosynthesis consumes TIC and through phytoplankton mortality releases TOC to the water, and exchange of $CO_2$ through the water-air interface. To evaluate the model's responsiveness to these processes, we varied the associated parameters by ±20% around their calibrated values. This approach allows us to assess the robustness of the model and identify which processes exert the greatest influence on carbon loss from the aquatic system.

### 3. Study area and data description

### 3.1. Description of the Study site - Vantaanjoki catchment

The Vantaanjoki catchment, covering an area of 1680 km$^2$, is located in southern Finland, just north of the city of Helsinki and includes Tuusulanjärvi lake (Fig. 3a&b). The river flows southwards to the Gulf of Finland. Forests on mineral soils, forests on peat soils, fields and lakes cover 50%, 7%, 22% and 3% of the catchment, respectively, with the remaining 18% being covered by urban areas (Corine Land Cover 2018 and Peatland drainage status (Syke), Fig. 3b). The soil types are distributed



between north and south with northern areas characterized by sand/till/silt/organic mostly covered by forests while the south

is defined by clay overlay allowing for more agriculture to take place. Overall, clay is the main soil type (37%) followed by

rock, sand and till (22%, 12% and 11%, respectively) while organic soils and silt both cover 8% of the total catchment area

(Fig. 3c). The average rainfall over the 10 year period 2014-2023 is 720mm. In southern Finland hydrology is characterized

by the highest flow peaks occurring in spring during snowmelt and in autumn when rainfall is more abundant.

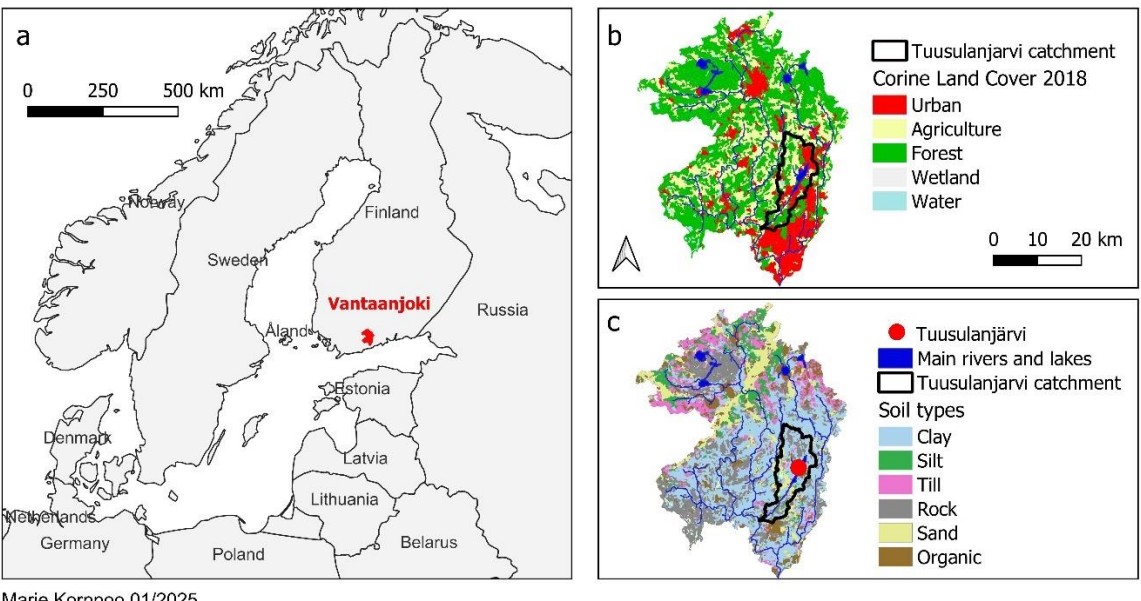

Marie Korppoo 01/2025
Source: b- Corine Land Cover - Finnish Environment Institute (Syke), LUKE, MAVI, LIVI, DVV, EU, NLS 01/2017
Source: c- Soil types: Geological Survey of Finland (GTK) and Natural Resources Institute Finland (LUKE)


**Figure 3: a- Vantaanjoki catchment in red in the European map; b- land cover (Corine Land Cover, 2018) in Vantaanjoki catchment; c- soil types in Vantaanjoki catchment**

Tuusulanjärvi catchment area is 91km$^2$ of which 50% is covered by clay, 21% by rock and 12 % by till (Fig. 3c). Lake coverage

accounts for another 9% of the area, while forest (33%), agriculture (31%) and urban (27%) areas are equally distributed (Fig.

3b). Tuusulanjärvi Lake itself covers 5.9 km$^2$ and has a volume of 18 million m$^3$ with a retention time of over 200 days and is

the largest lake in the Vantaanjoki catchment. The lake is hypereutrophic (Schönach et al., 2018).

**3.2 Monitoring data**

Mainly nine discharge as well as ten lake water level observation points are used for the runoff parameter calibration of the

hydrological model in Vantaanjoki catchment (Table 1). Since there are no large lakes in the catchment, the discharge

observation points get more weight in the runoff parameter calibration. The calibration is done by minimizing the difference





between daily simulated and observed discharge in all points simultaneously. The points with larger discharge get higher weight in the optimization function.

Both TOC observations and TOC calculated from $COD_{Mn}$ observations were used in model calibration and validation. TOC data were estimated from $COD_{Mn}$ data that correlates well with TOC in Finland (Kortelainen, 1993). As TOC data for rivers and lakes are scarce and $COD_{Mn}$ have been more often analyzed, this approach enables to use larger data sets and improved model performance. Equation (13) is fitted for Vantaanjoki catchment with CODMn and TOC observations from 1990-2022.

$$TOC = 0.797 \times COD_{Mn} + 1.076, r^2 = 0.89, n = 4193 \qquad (13)$$

TIC observations (mgC L-1) were scarce all over Finland and thus alkalinity (Alk in mmol L-1) measurements in lake waters were used as a proxy using the expression (Rantakari and Kortelainen, 2008 with an $r^2$=0.95, Eq. (14)):

$$TIC = (0.996 * Alk * 1000 + 19.3) * \left(\frac{12}{1000}\right) \qquad (14)$$

TOC and TIC are measured on unfiltered samples in Finland, while DOC and DIC are very rarely sampled. Water quality observation points monitored regularly in the catchment are used in the calibration of the Vemala model. Vantaa 4,2 6040 is the most frequently monitored water quality observation point downstream of the catchment with 15 or more observations per year (Fig.5a). Based on the observations available in the Hertta data management system of Syke; over the period 1990–2023, the mean (±standard deviation) total phosphorus, total nitrogen, TOC and TIC concentrations were 102 µg L$^{-1}$ (±61), 2.3 mg L$^{-1}$ (±0.9), 12.3 mg L$^{-1}$ (±3.5) and 10 mg L$^{-1}$ (±2.7) respectively and for alkalinity and pH the averages were 0.79 (±0.23) and 7.27 (±0.28). The ecological status of Vantaanjoki and Tuusulanjärvi is considered satisfactory (Syke).

Additionally, samples were collected from the main tributaries and outlet of the Tuusulanjärvi lake during 2023. The samples were analysed at the MetropoliLab Oy for total organic carbon (SFS-EN 1484:1997), total inorganic carbon (SFS-EN 1484:1997), pH (SFS 3021:1979) and alkalinity (VYH87).

## 4. Results

### 4.1 River basin scale results

#### 4.1.1 Discharge

Table 1 shows the discharge observation points used for model calibration, validation and NSE values for both periods. NSE for calibration period varies between 0.68-0.97, and for validation period between 0.87-0.97. The model can simulate reasonably well observation points on small and medium-sized catchment areas. Figure 4a shows the daily simulated and observed discharge at Vantaanjoki Oulunkylä station (2004-2023) with a NSE=0.97 for the calibration period (Fig 4a). Water level observations of unregulated lakes are used to calibrate the lake rating curve parameters. For regulated lakes, such as Tuusulanjärvi, outflow (Fig. 4b) and water level measurements, provide an estimate of 'observed' inflow to the lake, which is used in runoff parameter calibration.



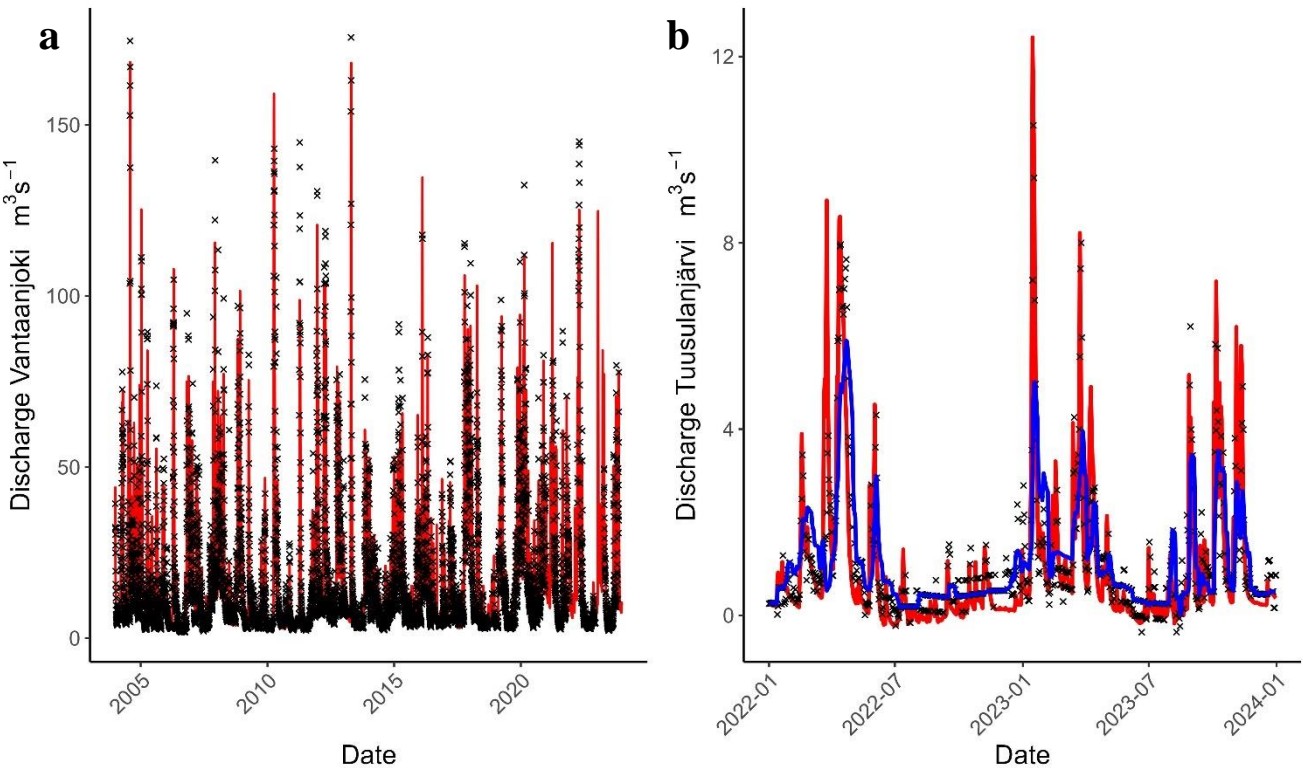

**Figure 4. a- Simulated (red line) and observed (black cross) discharges at Vantaanjoki outlet for calibration period 2004 – 2023. b- Observed inflow (black cross), simulated inflow (red line) and simulated outflow (blue line) of Tuusulanjärvi for calibration period 2022 – 2023.**








Table 1. Discharge observations used for model calibration and validation, and NSE values for calibration and validation periods

| Nr. | Name of the station | Period | Catchment area, km2 | NSE. Calibration 2004-2023 | NSE. Validation 1990-2003 |
|---|---|---|---|---|---|
| 1 | 2101700 Oulunkylä | 1990-2023 | 1688.52 | 0.97 | 0.97 |
| 2 | 2101220 Myllymäki | 1990-2023 | 1232.2 | 0.96 | 0.96 |
| 3 | 2101500 Vantaa,Ylikylä | 2001-2023 | 556.36 | 0.88 | 0.88 |
| 4 | 2101520 Hanala | 1990-2023 | 313.38 | 0.92 | 0.91 |
| 5 | 2104900 Lepsämänjoki lm | 1996-2023 | 214.07 | 0.93 | 0.93 |
| 6 | 2105400 Tuusulanjoki, Myllyky | 2005-2010 | 125.74 | 0.96 | - |
| 7 | 2100946 Sandbacka | 1990-2023 | 57.38 | 0.85 | 0.87 |
| 8 | 2100100 Lepsämänjoki anturi | 2005-2018 | 23.36 | 0.73 | - |
| 9 | 2100101 Laurinoja | 2010-2015 | 1.17 | 0.68 | - |

**4.1.2. Total organic carbon**

TOC concentrations in Finland using both TOC observations and $COD_{Mn}$ observations as a proxy (Eq. (13)) range between 0.1-480 mg $L^{-1}$, with a mean and median of 12.5 and 9.8 mgC $L^{-1}$ respectively, based on 106512 observations available after 2000 from the Vesla database (Syke). The TOC concentrations in Vantaanjoki (13.1 mgC $L^{-1}$) and Tuusulanjärvi (9.8 mgC $L^{-1}$) are around the national mean and median respectively.

Vemala TOC model simulates the source apportionment of the TOC loading as well as spatial distribution of gross TOC
loading for each small brook catchment. Figure 5a shows the simulated TOC specific loading spatial distribution, which depends on soil and land use classes. There are higher TOC loading areas in Vantaanjoki upstream sub-catchments, where relatively more peat soils are located. Simulated TOC specific loading from forests on drained peat soils is 9.2 tC $km^{-2}$ $yr^{-1}$, natural peat soils: 7.8 tC $km^{-2}$ $yr^{-1}$, forests on mineral soils: 3.6 tC $km^{-2}$ $yr^{-1}$ and agriculture on mineral soils: 3.0 tC $km^{-2}$ $yr^{-1}$. Even if the highest TOC specific loading is from peat soils, the highest source of TOC is forests on mineral soils (62%) as they
cover the largest area in Vantaanjoki catchment. For the TOC daily loads at the Vantaanjoki outlet observation point, NSE was 0.65 for the calibration period (2004-2023), and 0.79 for the validation period (1990-2003) (Table 3).





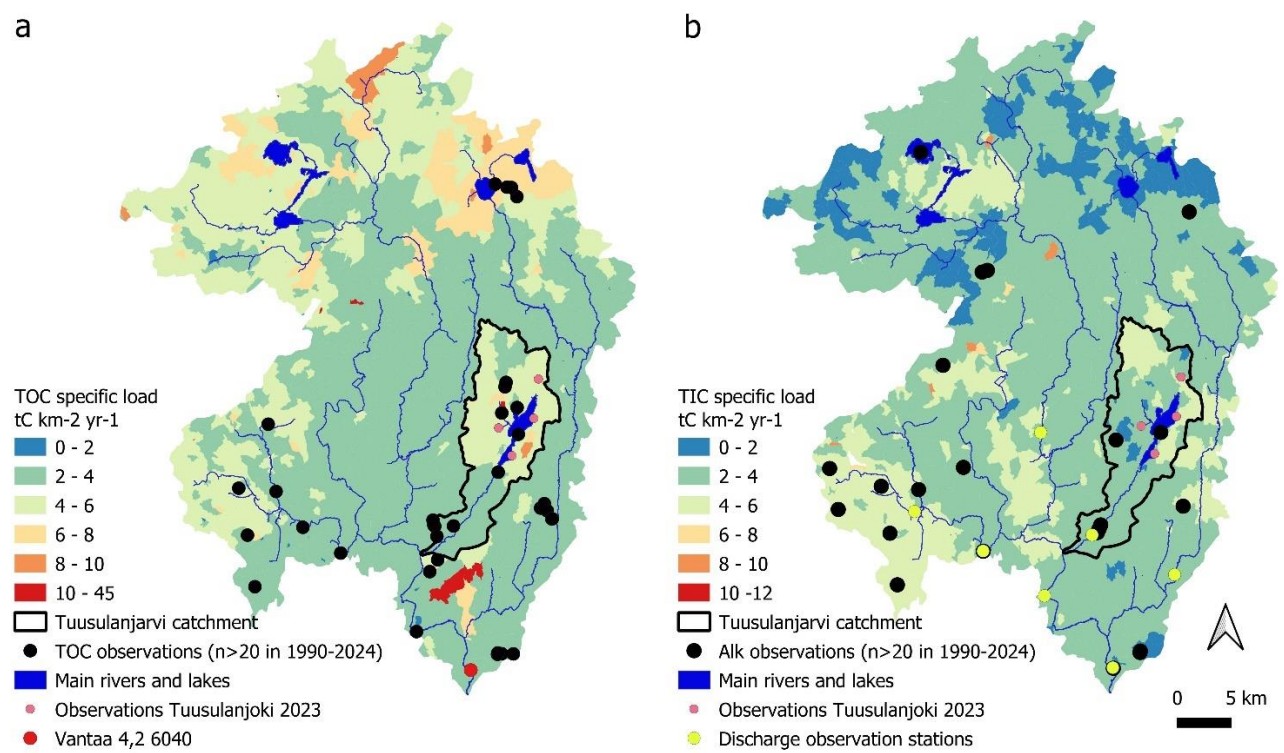

**Figure 5. a- Simulated TOC specific loading (tC km⁻² yr⁻¹) in Vantaanjoki, and TOC observation points, > 20**
**observations for the period 1990-2023 b- Simulated TIC specific loading (tC km⁻² yr⁻¹) in Vantaanjoki, and alkalinity**
**observation points, > 20 observations for the period 1990-2023. Source: WSFS-Vemala / Finnish Environment Institute**
**(Syke)**

### 4.1.3. Total inorganic carbon

TIC concentrations in Finland using alkalinity observations as a proxy (Eq. (14)) range between 0.1-85 mgC L⁻¹, with a mean
and median of 3.9 and 2.4 mgC L⁻¹ respectively, using 23508 observations available after 2000 from Vesla. The TIC
concentrations in Vantaanjoki (9.7 mgC L⁻¹) and Tuusulanjärvi (8.5 mgC L⁻¹) are more than double the national mean.
Vemala simulates rock weathering and the resulting alkalinity, and thus TIC (Eq. (14)), from various soil types. There are 12
observation points for alkalinity in Vantaanjoki with more than 20 observations over the period 1990-2023. The highest
alkalinity is calibrated as coming from rock and clay soils at runoff higher than 0.5 mm, while for lower runoff, sand and till
are contributing the most to alkalinity (Table 2). Figure 5b shows the spatial distribution of simulated TIC specific loading in
Vantaanjoki catchment ranging from 1-7 tC km⁻² yr⁻¹ depending on soil types.





Table 2: Calibrated parameters for the simulation of alkalinity in the Vantaanjoki catchment

| Soil type | Calibrated range mmol L⁻¹ | Γ | ε |
|---|---|---|---|
| Sand | 0.6 ±20% | 0.513 | -0.68 |
| Silt | 0.75 ±20% | 0.6 | -0.26 |
| Clay | 1.25 ±20% | 1.02 | -0.24 |
| Organic | 0.12 ±20% | 0.1 | -0.26 |
| Till | 0.7 ±20% | 0.745 | -0.67 |
| Rock | 1.5 ±20% | 1.29 | -0.25 |


Table 3: Nash-Sutcliffe coefficient of the carbon and nutrient loading at the outlet of Tuusulanjärvi (Tuusulanjärvi luusua 1) over the calibration (2004-2023) and validation (1990-2003) periods with total organic carbon (TOC in mgC L⁻¹), total inorganic carbon in the water (TIC in mgC L⁻¹)

| Observation point | Period | TOC (tC d⁻¹) | TIC (tC d⁻¹) | Alk | pH | TP (kgP d⁻¹) | TN (tN d⁻¹) |
|---|---|---|---|---|---|---|---|
| Tuusulanjärvi luusua1 | 1990-2003 | | | | | 0.93 (n=163) | 0.85 (n=162) |
| | 2004-2023 | 0.94 (n=26) | 0.80 (n=8) | 0.95 (n=7) | 0.99 (n=84) | 0.84(n=204) | 0.74 (n=188) |
| Vantaanjoki 4,2 6040 | 1990-2003 | 0.79 (n=193) | 0.77 (n=12) | -0.50 (n=197) | 0.98 (n=678) | 0.85 (n=245) | 0.71 (n=245) |
| | 2004-2023 | 0.65 (n=242) | 0.63 (n=199) | 0.60 (n=241) | 0.99 (n=436) | 0.74 (n=397) | 0.70 (n=436) |

For the alkalinity daily loads at the Tuusulanjärvi outlet observation point, NSE was 0.95 for the calibration period (2004-2023), and for Vantaanjoki outlet NSE was 0.60 for the calibration period and was negative for the validation period (Table 3). The discrepancy in the results of alkalinity is explained by an increased alkalinity in Vantaanjoki over the calibration period with an average alkalinity of 0.8 mmol L⁻¹ (Fig. A3) compared to the validation period with an average of 0.62 mmol L⁻¹.

### 4.1.4. River basin carbon budget

The catchment annual loads of TOC can be compared to estimates derived from the monitoring data and an averaging method for the period 1991-2020 (Helcom PLC database). The average net annual loading at the outlet of Vantaanjoki over the period 1991-2020 was 5974 tC yr⁻¹ in Vemala and 7098 tC yr⁻¹ in the estimation. Vemala underestimated the annual TOC loading estimates by on average -16% over the period with a range from -33% to +25% with the largest underestimation in the wet years 2012 and 2017. In 2012, only two samples have been recorded in October and November with high values (28 and 22 365    mgC L⁻¹). These peaks in concentrations are underestimated in Vemala, however, it is possible that the averaging method overestimates the loading when using the interpolation method with too few values for the autumn loading.



The annual terrestrial average TC loads (gross loading) for the Vantaanjoki catchment were 11210 tC yr[-1] and 12870 tC yr[-1] respectively for the validation and calibration period. The annual average loading to the Sea (net loading) for TC was 8730 tC

yr[-1] and 9675 tC yr[-1] respectively for the validation and calibration periods with about 66% being TOC loading and 34% TIC loading. The model simulated a retention/release of carbon through the river and lake network of 2480 tC yr[-1] and 2715 tC yr[-1] respectively for the validation and calibration period or about 22% of the terrestrial total carbon reaching the aquatic ecosystem. About half of the loss occurs in the lakes of the catchment (1030 tC yr[-1]) that cover a total area of 37 km$^2$ (sizes range from 0.001-5.9km$^2$). If we consider a river network area of 6.5km$^2$, based on the geometry of river stretches, defined in

the Finnish basemap (Metadata: Shoreline10 from Syke and MML) with a discharge larger than 0.05 m$^3$s$^{-1}$ (WSFS-Vemala, Syke), the resulting $CO_2$ emission rate is between 223-260 gC m$^{-2}$ yr$^{-1}$ in the river network itself.

## 4.2 Lake processes

### 4.2.1. Tuusulanjärvi tributaries

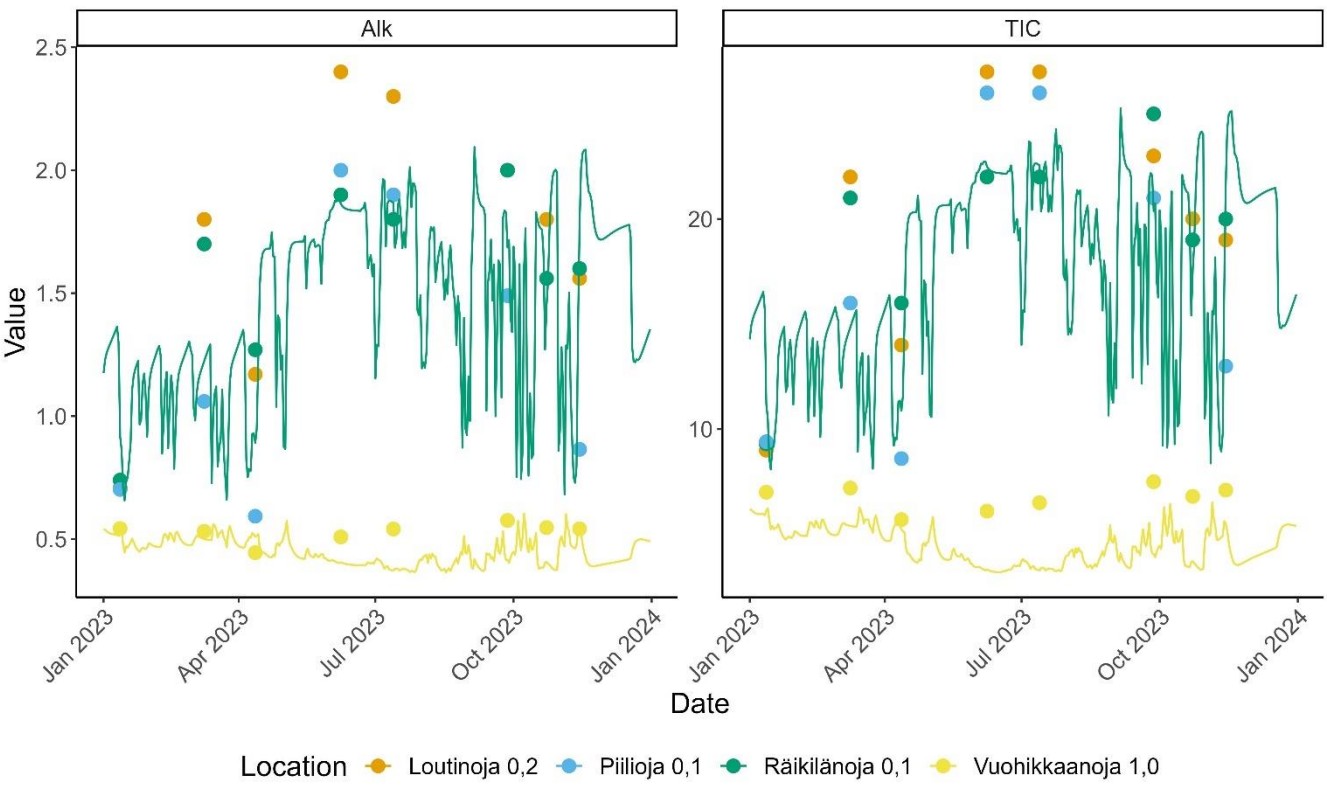


**Figure 6: Observations (dots) and Vemala simulations (lines) of water quality variables in the rivers upstream Tuusulanjärvi in 2023 (Alk=alkalinity in mmol L$^{-1}$, TIC=total inorganic carbon in mg L$^{-1}$), monitored by the ELY Centres (Centres for Economic Development, Transport and the Environment in Finland)**



The alkalinity in the various tributaries monitored over the year 2023 can be divided into two groups, one presenting a lower
alkalinity around 0.5-0.7 mmol L$^{-1}$ throughout the year and one presenting an alkalinity increasing during the summer months
up to around 2 mmol L$^{-1}$ or TIC concentrations up to 27 mgC L$^{-1}$ (Fig. 6). High TIC summer concentrations are related to a
large proportion of clay and rock in the catchments with up to 98% in Räikilänoja 0.1 (Table A2). Vuohikkaanoja 1.0 on the
other hand, which has only 40% of rock and up to 29% of sand presents low alkalinity and TIC concentrations throughout the
year. Vemala results represent well the variability of TIC seasonal concentrations based on soil types in the upstream
catchments (Fig. 6).

### 4.2.2 Water quality Tuusulanjärvi

Tuusulanjärvi is a hypereutrophic lake presenting chlorophyll-a values up to a 100 µg L$^{-1}$ during the summer (Vesla). Vemala
simulates well carbon (TOC, TIC, alkalinity, pH), nutrients (TN, nitrate (NO$_3$), TP, phosphate (PO$_4$)) and Chlorophyll-a (Chla)
concentrations in Tuusulanjärvi lake compared with observations (Tuusulanjärvi syvänne 1-2m depth, Fig. 7) over the
calibration period with both the inter and intra annual levels well represented. The annual increase in TIC concentrations in
the spring can be associated with the accumulation of CO$_2$ in the water over the winter period due to ice cover preventing the
exchange of CO$_2$ gas at the water-air interface. However, as noted in the simulation of alkalinity at the outlet of the Vantaanjoki
catchment, the increase in alkalinity in the later period of the calibration is visible from an average of 0.65 (2004-2012) to 0.75
(2013-2023). This increase cannot be explained by rainfall and discharge alone as the model simulates well the discharge in
the catchment over both periods. A slight increase in pH can also be noted in the Tuusulanjärvi observations over the validation
period, possibly due to the recovery to acidification or changes in soil weathering rates, unlike in the modelled results of pH.
The model also simulates well the carbon and nutrients loads at the outlet of Tuusulanjärvi (Tuusulanjärvi luusua 1, table 3)
over the calibration and validation periods with NSE values higher than 0.74 for TOC, TIC, alkalinity, pH, TP and TN (Table
3).



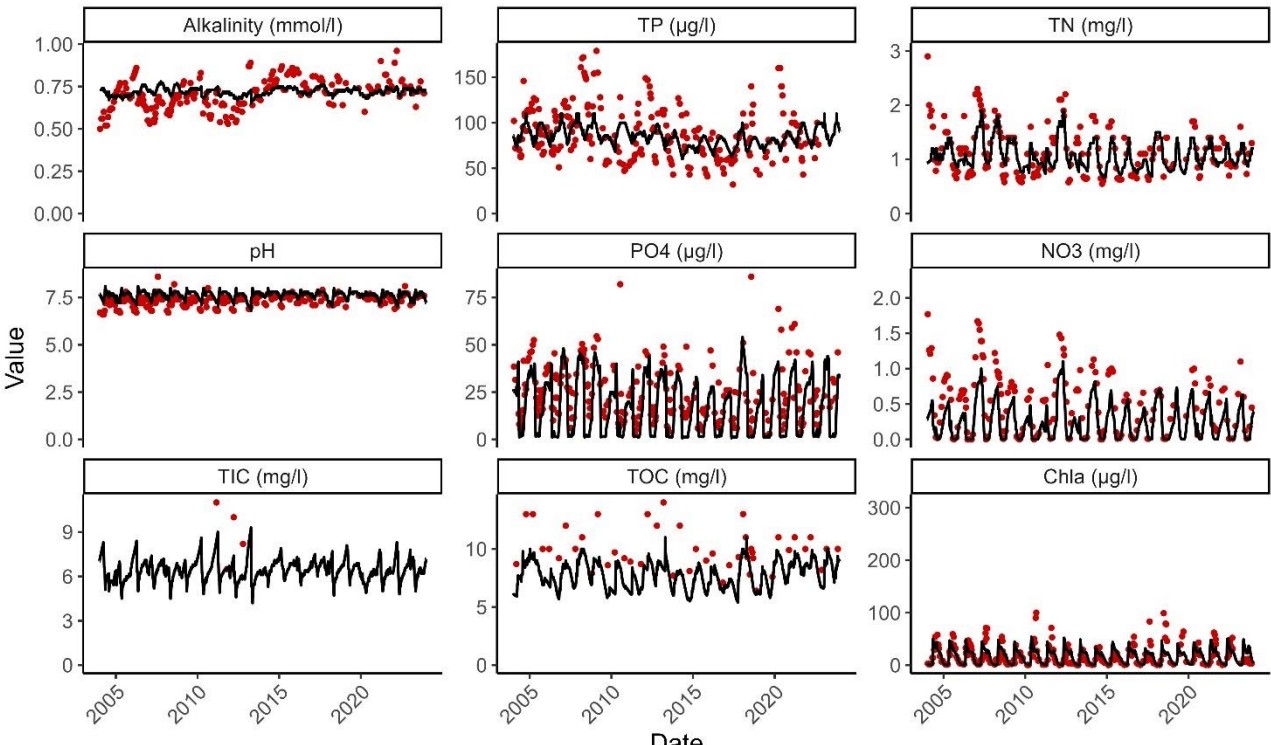

**Figure 7: Water quality in Tuusulanjärvi over the period 2004-2023 with observations (red dots) and simulations (black line) for alkalinity (mmol L$^{-1}$), pH, total inorganic carbon (TIC, mg L$^{-1}$), total phosphorus (TP, μg L$^{-1}$), phosphate (PO$_4$, μg L$^{-1}$), total organic carbon (TOC, mg L$^{-1}$), total nitrogen (TN, mg L$^{-1}$), nitrate (NO$_3$, mg L$^{-1}$) and phytoplankton (Chla, μg L$^{-1}$).**

The average annual $CO_2$ emissions (22-27 gC m$^{-2}$ yr$^{-1}$) simulated by Vemala have increased between 1990-2003 and 2004-2023 along with an increase of $CO_2$ dissolved in the water from 537 mgC m$^{-3}$ to 547 mgC m$^{-3}$ (Table 4). This result can mainly be explained by higher water temperature, lower pH and higher wind speeds in the second period with a k$_{600}$ increasing from 6.7 to 7.1 cm h$^{-1}$. The increase in the atmospheric $CO_2$ from 362 to 400 ppm (Eq. (11)) does not counterbalance the effect of wind, temperature and pH on the $CO_2$ emissions between the periods (Table 4).



Table 4: Annual average $CO_2$ in the water ($CO_{2aq}$ in ppm), $CO_2$ at saturation ($CO_{2sat}$ in ppm), total inorganic carbon in the water (TIC in mgC $L^{-1}$), total organic carbon in the water (TOC in mgC $L^{-1}$), $CO_2$ transfer velocity at water temperature ($k_{CO2}$ cm $h^{-1}$) and at 20°C ($k_{600}$ in cm $h^{-1}$), average daily (F in mgC $m^{-2}$ $d^{-1}$) and annual (F in gC $m^{-2}$ $yr^{-1}$) $CO_2$ emissions to the atmosphere, atmospheric $CO_2$ ($CO_{2atm}$ in ppm) and pH averaged over the calibration period 2004-2023 and validation period 1990-2003 as simulated by Vemala in Tuusulanjärvi.

| Period | $CO_{2aq}$ (mg $m^{-3}$) | $CO_{2sat}$ (mg $m^{-3}$) | TIC (mg $L^{-1}$) | TOC (mg $L^{-1}$) | $k_{CO2}$ (cm $h^{-1}$) | $k_{600}$ (cm $h^{-1}$) | F (mgC $m^{-2}$ $d^{-1}$) | F (gC $m^{-2}$ $yr^{-1}$) | $CO_{2atm}$ (ppm) | pH | Water temperature (°C) |
|---|---|---|---|---|---|---|---|---|---|---|---|
| 1990-2003 | 537 (400-628) | 272 (258-285) | 6.7 (6.3-7.3) | 7.3 (5.4-8.3) | 3.1 (2.7-3.7) | 6.7 (6.1-7.4) | 61 (43-80) | 22 (16-29) | 361 (347-375) | 7.7 (7.6-7.8) | 6.7 (5.9-7.3) |
| 2004-2023 | 547 (382-778) | 297 (281-314) | 6.6 (6.0-7.0) | 7.7 (6.7-8.7) | 3.7 (2.9-4.7) | 7.1 (6.1-7.8) | 73 (53-94) | 27 (19-34) | 400 (377-422) | 7.6 (7.6-7.7) | 7.2 (6.2-7.9) |

### 4.2.3 Carbon emissions and long-term sedimentation

A simplified representation of the simulated average annual total carbon budget for the Tuusulanjärvi lake over the period 1990-2023 (Fig. 8) shows that most of the carbon (73%) is flowing downstream of the lake (474 tC $yr^{-1}$). However, a significant amount of carbon is released to the atmosphere annually (net emissions: 23% or 147 tC $yr^{-1}$ or 25 gC $m^{-2}$ $yr^{-1}$) while a smaller portion is sedimented on the long-term (3% or 20 tC $yr^{-1}$ or 3 gC $m^{-2}$ $yr^{-1}$). From the total carbon losses, TOC losses (108 tC $yr^{-1}$) represent 30% of the incoming TOC loading while TIC losses (65 tC $yr^{-1}$) represent 24% of the TIC loading to the lake. Primary production (PP) in the lake represents 200 tC $yr^{-1}$, which is larger than the amount of $CO_2$ net emissions from the lake. The PP is either sedimented (82 tC $yr^{-1}$) or returned to the water as TOC (respiration, 105 tC $yr^{-1}$) or exported as phytoplankton downstream (8 tC $yr^{-1}$) before being mineralised back into TIC.





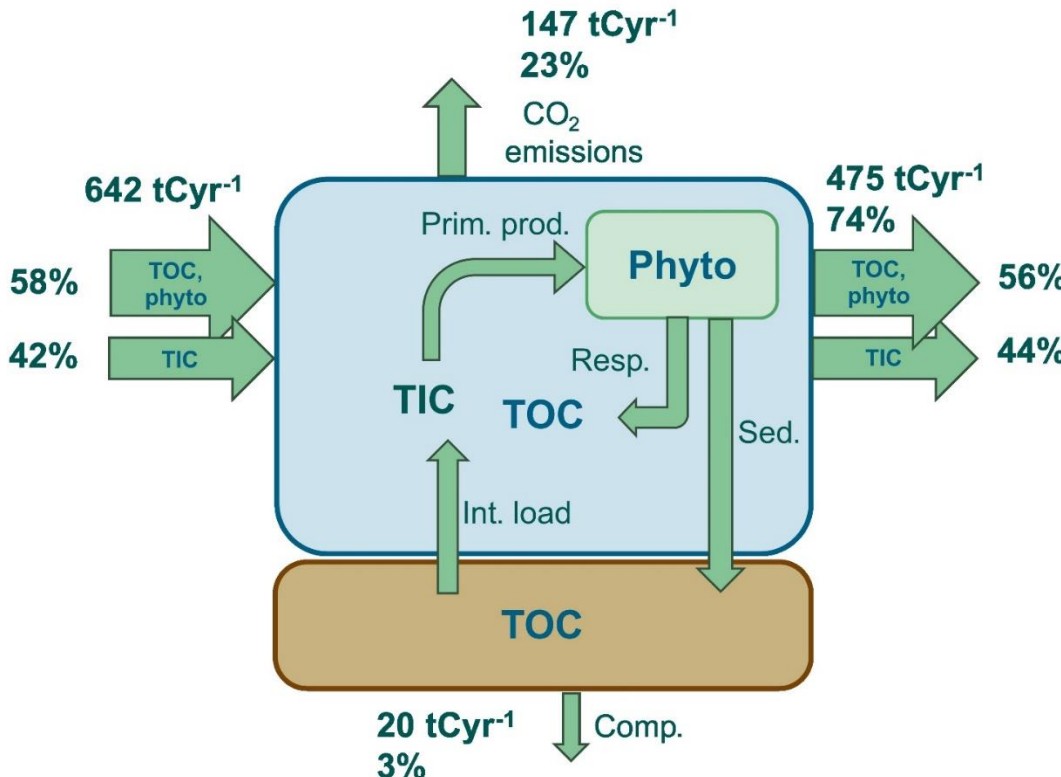

**Figure 8: Simulated Tuusulanjärvi carbon budget (WSFS-Vemala) over the period 1990-2023 with TIC (total inorganic carbon), TOC (total organic carbon), phyto (phytoplankton), Prim.prod. (primary production), Resp. (respiration), Sed. (sedimentation), Comp. (burial), Int. load (internal loading)**


### 4.2.4 Sensitivity analysis

The simulations of Tuusulanjärvi water quality for the period 2004-2023 show low sensitivity to a ±20% change in the $CO_2$ transfer coefficient across the water-atmosphere interface. The resulting variations in TIC ranged from –0.6% to +1.3%, in TOC by approximately ±0.1%, and in $CO_2$ emissions by 0.3–2% (Figure 9). Similarly, a ±20% change in phytoplankton growth
rates had minimal impact on model outputs: TIC varied between 0.0% and +0.3%, TOC between –0.5% and +0.5%, and $CO_2$ emissions increased by +0.8% to +2%. However, the new model is more sensitive to changes in mineralisation rates. A ±20% variation in the mineralisation rate resulted in $CO_2$ emissions changing by –7% to +8%. TOC concentrations were even more responsive, showing an opposite effect with changes of –8% to +10%, while TIC concentrations were moderately affected, varying between –3.5% and +3%.


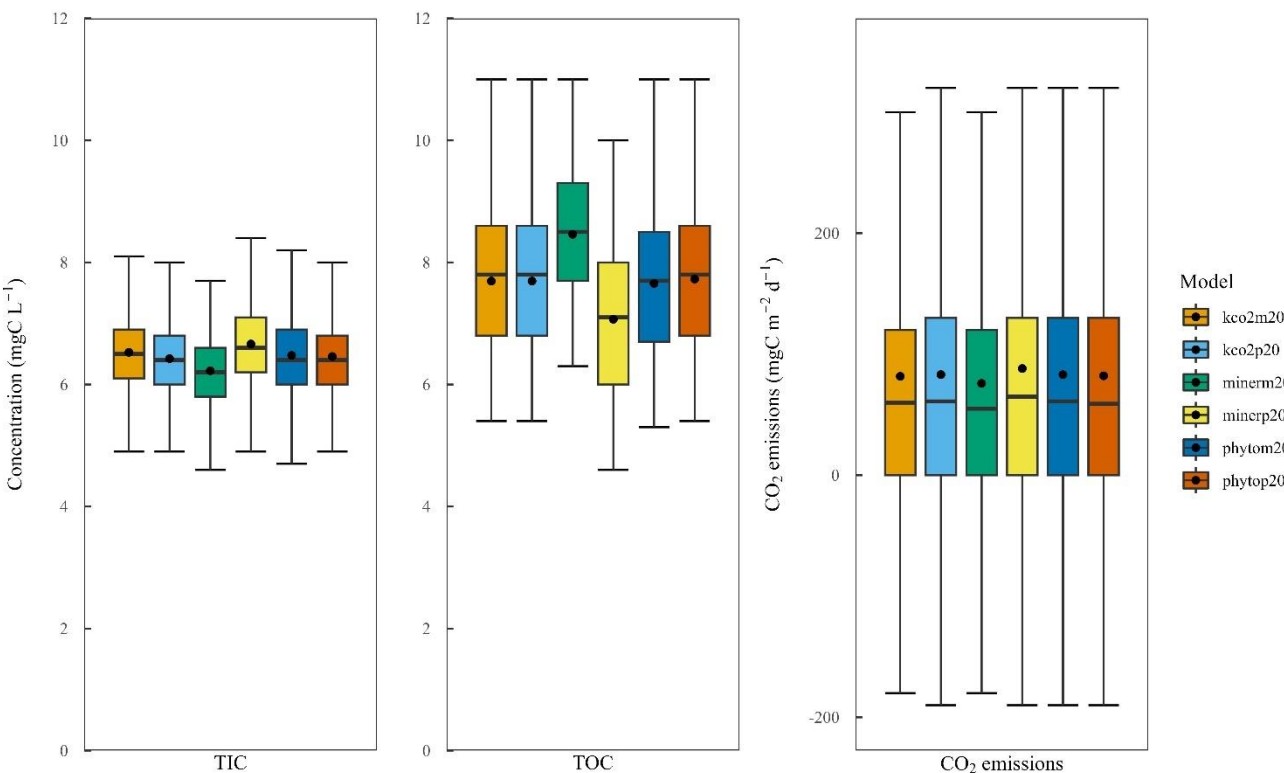

**Figure 9: Sensitivity analysis of daily total inorganic carbon (TIC in mgC L$^{-1}$) and total organic carbon (TOC in mgC L$^{-1}$) concentrations in the water, and daily CO$_2$ emissions to/from the atmosphere (F in mgC m$^{-2}$ d$^{-1}$) to an** 460 **increase/decrease of the mineralisation rates (minerp20/minerm20), of the phytoplankton growth (phytop20/phytom20) and of the CO$_2$ exchange rate (kco2p20/kco2m20) by 20% over the period 2004-2023. The mean is represented by the black dot.**

## 5. Discussion

A long-term goal for Vemala model development is to be able to simulate TOC and TIC leaching and transport at the national 465 scale, involving model application on all types of land use/soil classes and over a wide range of climatic conditions. This is important especially in a lake abundant country such as Finland, where carbon processes in a large number of lakes will be affected by changing climate and human actions including the implementation of water and climate related policies. Subsequent changes in carbon release and sedimentation in inland waters could then be added to estimates of the national carbon balance and formal GHG reporting processes.



## 5.1 TOC


In Vantaanjoki, the simulated TOC specific loading for different land use/soil classes compares well with values reported in the literature. Vemala simulates 3 tC km$^{-2}$ yr$^{-1}$ TOC load from agriculture on mineral soils, which is in line with Manninen et al. (2018) who reported 2.5–5.2 tC km$^{-2}$ yr$^{-1}$ DOC load from cultivated fields on mineral soils. The DOC load has been reported to increase with increasing topsoil carbon content (Manninen et al., 2018). Vemala simulates a TOC load of 3.6 tC

km$^{-2}$ yr$^{-1}$ from forests on mineral soils, which corresponds with Rantakari et al. (2010) findings 2.3 – 4.0 tC km$^{-2}$ yr$^{-1}$ of TOC load from forests on upland soils. TOC load from peat soils varies over a wide range depending on whether it is natural or drained, nutrient rich or oligotrophic and of the thermal gradient from south to north. Mean simulated TOC load for Vantaanjoki catchment from natural and drained peat soils were 7.8 and 9.2 tC km$^{-2}$ yr$^{-1}$ respectively, which are lower than 10.1 and 16.8 tC km$^{-2}$ yr$^{-1}$ calculated by export coefficient equations based on temperature sums for Vantaanjoki (Finer et al.,

480 2021).

There is an increasing trend in both simulated and observed TOC concentrations in Tuusulanjärvi for the period 1990-2023. Observed mean TOC concentrations in 1990's has increased by 16% for the decade 2010-2020. The increasing TOC trends in various waterbody types, streams, lakes, rivers and coastal waters, has also been reported in Finland by Räike et al. (2024). According to Räike et al. (2024), the main reasons for increasing trends of TOC transport to waterbodies are decrease in acid

sulphate deposition, increase in temperature, runoff, tree biomass and peat soil drainage.

## 5.2 TIC and alkalinity

Vemala performs well in simulating the loads of TIC, alkalinity and pH over both the calibration and validation periods with NSE higher than 0.6 at the outlets of Tuusulanjärvi and Vantaanjoki. There is however, one exception, with the NSE calculated

for alkalinity over the validation period being negative (-0.5). In Tuusulanjärvi and Vantaanjoki, the observed alkalinity increase between the calibration and validation periods can be partly explained by an increased buffer capacity of freshwaters following a decrease in acid deposition (e.g. Vuorenmaa, 2007, de Wit et al., 2016) from the late 1980s onwards. Sulphate deposition load declined significantly by 60-70% over the period 1986-2003 in southern Finland (Vuorenmaa, 2007). In addition, soil weathering rates are sensitive to increasing temperatures, with predicted increases in alkalinity and pH of Finnish

surface waters under climate change scenarios (Aherne et al., 2011). The Vemala model simulates alkalinity as a conservative tracer and therefore does not consider the above secular changes due to acid deposition and temperature. Moreover, it omits the impact of nitrification, denitrification and other processes affecting alkalinity as implemented in other carbon models (e.g. Marescaux et al., 2020). This is a simplification that will be addressed in future development of the model.

The TIC loading at the outlet of the catchment is well simulated, although TIC concentrations overall are underestimated in

Vemala. In Tuusulanjärvi, some of the tributaries are presenting an increased TIC concentration in the summer up to 27 mgC L$^{-1}$, while others present a low concentration throughout the year. In our model, TIC input loading is based on soil types with



a higher concentration coming from bedrock and clay soils as analysed from well waters in Finland (Korkka-Niemi, 2001) rather than from coarse soils (sand and silt). Elevated summer concentrations can be explained by elevated groundwater flow or point sources from urban areas or wastewater treatment plants. Point sources of TIC are not included in this model nor is

land use at this stage of the model development. Runoff from agricultural land may increase DIC loading compared to forested land due to liming. In Sweden, Humborg et al. (2010) estimated that 4% of the carbonate and bicarbonate export to the sea in Sweden was from pollution, mainly from liming of agricultural land, which covers 8% of Sweden. However, it is difficult to assess the extent of the liming effect on agricultural land on the stream water pH and thus inorganic C dissolved in the water (Huotari et al., 2013). Overall, the simplified representation of TIC loading based on soil types and runoff in Vemala represents

well the variability of TIC loading between different catchments as well as the intra-annual variability of TIC concentrations, even though the overall loading is underestimated.

### 5.3 $CO_2$ emissions

The simulation of the transfer velocity of $CO_2$ across the water-air interface, represented by $k_{600}$, is crucial in the calculation of $CO_2$ emissions and requires the introduction of physical/topographic features like wind speed and water velocity to reduce

the uncertainty associated with $k_{600}$ estimates at the catchment scale (Alin et al., 2011, Rocher-Ros et al., 2019). In this study, $k_{600}$ was defined separately for lakes and rivers and correlated respectively to wind speed (Jonsson et al., 2008) and water velocity (Alin et al., 2011). Average values of $k_{600}$ in Tuusulanjärvi over both periods ($k_{600}$=6.1-7.8 cm h$^{-1}$ annual averages) are comparable to other measurements from lakes like Heiskanen et al. (2014) who calculated an average $k_{600}$ of 7cm h$^{-1}$ in a small boreal lake during autumn. Alin et al. (2011) presented a $k_{600}$ range between 5-55 cm h$^{-1}$ in rivers with a water velocity

between 0.05-1 m s$^{-1}$. In Vantaanjoki, the average water velocity ranges between 0.02-0.35 m s$^{-1}$ over the period 1990-2023 (WSFS-Vemala, Syke), resulting in a range of $k_{600}$ between 14-26 cm h$^{-1}$. This is below the range used to characterise rivers in Sweden ($k_{600}$ = 26-64 cm h$^{-1}$, Humborg et al., 2010), however, the velocity in the Swedish streams was much higher, ranging from 1-4 m s$^{-1}$, which would explain our lower estimates of $k_{600}$.

The average annual $CO_2$ emissions rate for Tuusulanjärvi of 25 gC m$^{-2}$ yr$^{-1}$ is comparable to other studies, for instance, using

the equation related to open water rainfall from Rantakari & Kortelainen (2005) provides an average annual $CO_2$ emission rate of 24 gC m$^{-2}$ yr$^{-1}$. Our estimate also compares to other studies like the estimate, based on continuous $CO_2$ concentrations in the water (April 2005-October 2006), of 30-44 gC m$^{-2}$ yr$^{-1}$ from a meso-eutrophic, slightly acidic headwater lake in southern Finland (Huotari et al., 2009). However, our estimate is about half the estimate for its lake size class (1-10km$^2$) of 56 gC m$^{-2}$ yr$^{-1}$ but corresponds to a lake class with an area greater than 100 km$^2$ in the study from Kortelainen et al. (2006). Kortelainen

et al.'s study did not account for residence time, eutrophication or acidity levels that can considerably influence carbon emissions from lakes.

The overall river network carbon emissions of 223-260 gC m$^{-2}$ yr$^{-1}$ were low compared to calculations based on the size of the rivers like in Humborg et al. (2010: 473-3032 gC m$^{-2}$ yr$^{-1}$, depending on the Strahler stream order). This discrepancy can first be explained by the discrepancy in the average velocity of rivers between the two papers as we have presented with the $k_{600}$



values. These high emissions have also been reported by Rocher-Ros et al. (2019: median 1290 and mean 2790 $gCm^{-2}yr^{-1}$) or Raymond et al. (2012: mean 3358 $gC\ m^{-2}\ yr^{-1}$). However, in peat areas, lower $CO_2$ emissions have been reported like Rocher-Ros et al. (2019: 40 $gC\ m^{-2}\ yr^{-1}$) or Dinsmore et al. (2010: 39 $gC\ m^{-2}\ yr^{-1}$). TIC concentrations in Finnish waters can play a role in the lower estimates of $CO_2$ emissions. Indeed, in boreal environments DOC flux represents up to 80% of the carbon export to the Baltic Sea (Räike et al., 2016), while worldwide DIC export to the Sea is equivalent to DOC (Hope et al., 1994) and can

even represent up to 80% of the carbon flux in UK rivers (Jarvie et al., 2017). In boreal catchments characterised by non-carbonate bedrocks, inland waters are low in alkalinity and in inorganic carbon explaining the lower $CO_2$ emissions. Finally, the lower estimates of carbon emissions in the streams could also be explained by an underestimation of TIC in the water column. The underestimation of TIC is due to a simplification of the TIC loading, which is in this model based uniquely on soil types and not incorporating the added diffuse sources of TIC from agriculture, peat or urban areas, nor the point sources

like wastewater treatment plants.

## 5.4 Lake carbon budget

The older version of the Vemala model, simulating TOC only as a separate variable and including a retention term for both carbon emission and carbon sedimentation in the lake, simulated a TOC retention of 108 $tC\ yr^{-1}$ equivalent to 20% of the TOC loading to the lake (Vemala, Huttunen et al., 2016). This value is lower than the combined total carbon loss from the lake in

our new estimates (147 $tC\ yr^{-1}$) that includes not only carbon retention in the sediments but also carbon emissions to the atmosphere. Thus, over a third of the total carbon loss is omitted in the previous version of the model showing the importance of simulating TIC as well as TOC in carbon emission calculations. In this simulation, a large part of the total carbon is sedimented through phytoplankton sedimentation. However, only a relatively small fraction undergoes long-term burial (3% of TC), referred in this model as "compacted", thus most of the sedimented TOC is mineralised in the surface of the sediments

and returns to the water column as TIC. The capture of carbon in the sediments (3 $gC\ m^{-2}\ yr^{-1}$) is lower than in other lakes of similar sizes (1-10$km^2$: 7.2 $gC\ m^{-2}\ yr^{-1}$) according to Pajunen (2000). The carbon cycling in inland waters has significantly changed due to human activities (Tranvik et al., 2009) and this link should be further explored using this model. It would be important to expand this study to the national scale to get the total estimate of carbon balance and flows in the aquatic environment, taking into account the variety in characteristics and loading of lakes and rivers. This would improve currently

available estimates on GHG fluxes from Finnish surface waters, which are based on generalised emission coefficients (Holmberg et al., 2023). Evaluation of the regional-scale impacts of future changes in climate or land-use could also be improved (e.g. Forsius et al. 2017). Water residence times and nutrient concentrations are key predictors for organic carbon reactivity and $CO_2$ emissions from inland surface waters (Evans et al., 2017).

## 5.5 Sensitivity analysis

In the Vemala biogeochemical model, mineralisation is at the core of the carbon cycling in the water, which explains the sensitivity of the model to this process. The mineralisation process is simulated as affected by temperature following a sigmoid




function (Korppoo et al., 2017). Future climate change scenarios will thus have an impact on the simulated mineralisation dynamics in the aquatic ecosystem. Mineralisation not only influences carbon dynamics but also affects nutrient availability and phytoplankton growth, making it a key driver of biogeochemical interactions in the system. The co-dependency of various variables to the mineralisation rates allows for a reduction in the uncertainty of the calibrated parameters as mineralisation rates are calibrated to fit not only TOC but also organic nitrogen and phosphorus. This demonstrates the benefit of deterministic process-based models that rely on well documented enzyme reaction rates as well as the simulation of interconnected variables affected by the same processes in the aquatic ecosystem. The sensitivity analysis highlights the performance of the model to represent well all variables in the water column considering the sensitivity of each output to each calibrated parameter.

## 6. Conclusion

The development of the carbon processes in WSFS-Vemala model presents a significant advancement in simulating carbon dynamics, particularly in capturing both TOC and TIC processes and their contributions to carbon retention and emissions through a river/lake network. This improved representation of carbon cycling in Vemala, linked with phytoplankton growth and nutrient cycling, allows for the distinction between carbon losses to the atmosphere and long-term carbon storage in the sediments of inland waters. The model's ability to simulate TOC loading across various land use/soil types and TIC loading across soil types aligns closely with reported literature values, demonstrating its applicability in diverse climatic and geographical settings. Moreover, the addition of organic acids to the total alkalinity definition improved pH simulations and thus $CO_2$ emissions in acidic and organic rich waters like in Finland. The model's performance in estimating $CO_2$ emissions shows good correlations with established ranges for rivers and lakes. The inclusion of sedimentation and mineralization processes in the lake carbon budget underlines the necessity of accounting for both organic and inorganic pathways in carbon modelling.

Future development efforts for Vemala model should aim at applying the model to various land use/soil classes and over a wide range of climatic conditions. This is important especially in a lake abundant country such as Finland, where carbon processes in a large number of lakes will be affected by changing climate and human actions including the implementation of water and climate related policies. Overall, the enhanced Vemala model provides a robust foundation for understanding carbon cycling and supporting sustainable, integrated water resource management and climate change scenario assessments from sub-catchments to the national scale. Subsequent changes in carbon release and sedimentation in inland waters could then be added to estimates of the national carbon balance and formal GHG reporting processes.

## Author contribution

MK, IH and MH developed the model code, MK and JS designed the sampling campaign and MK and IH prepared the manuscript with contributions from all co-authors. The authors declare that they have no conflict of interest.



**Acknowledgements**

The development of the WSFS-Vemala carbon model has been funded by the Finnish Environment Institute (Syke) and the Ministry of Agriculture and Forestry of Finland (MMM, VN/28536/2020) through the Systeemihiili project and by the European union (NextGenerationEU and Water4all) through the Research Council of Finland projects: Green-Digi Basin: Green and digital transition in river basin management project (347703), Blue Lakes: digitizing the carbon sink potential of boreal lakes project (353316), C-NEUT: Evaluating integrated spatially explicit carbon-neutrality for boreal landscapes and regions (347848), Pluralakes: Co-creating pathways to desirable nature futures of temperate lakes (367832) and Festival: Final Ecosystem Services in Transition: Aquatic Ecosystem Services in Forested Landscapes (367822).

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





**Appendix A**

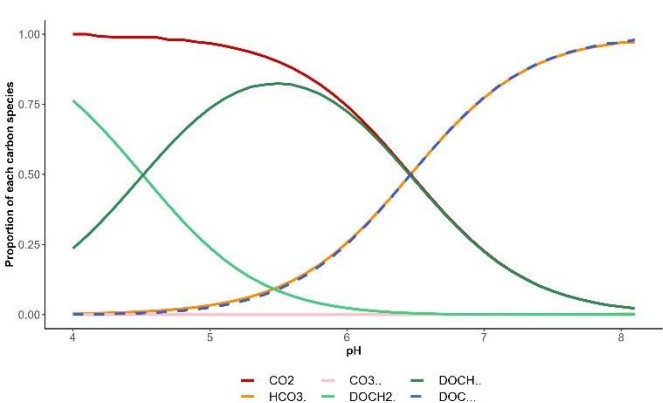

**Figure A1: Carbon species proportion for both dissolved organic carbon (DOCH$_2^-$ light green, DOCH$^{2-}$ green, and DOC$^{3-}$ dashed blue) and dissolved inorganic carbon (CO$_2$ red, HCO$_3^-$ orange, CO$_3^{2-}$ pink) depending on pH**

**pH model**

Solubility of $CO_2$ is defined by Weiss 1974:

$ln\,Ko = -60.2409 + 93.4517(100/T) + 23.3585\,ln\,(T/100)$ (1)

Where $K_0$ is expressed in moles kg$^{-1}$atm and T the temperature in Kelvin.

Alkalinity is defined as the ability of water to neutralize acids. The expression to calculate alkalinity in this paper using both TIC and TOC is given by:

$$Alk = [HCO_3^-] + 2[CO_3^{2-}] + [OH^-] - [H^+] + [DOCH_2^-] + 2[DOCH^{2-}] + 3[DOC^{3-}] \quad (2)$$

With,

$$[TIC] = [H_2CO_3] + [HCO_3^-] + [CO_3^{2-}] \quad\quad\quad (3)$$

$$[TOC] = [DOCH_3] + [DOCH_2^-] + [DOCH^{2-}] + [DOC^{3-}] \quad\quad (4)$$

With DOCH$_3$ omitted from the following equations as its process is outside the range of freshwater pH (pKa=3.04, Hruska et al., 2003).

The dissociation of water is determined by:

$$K_w = [OH^-][H^+] \quad\quad\quad\quad (5)$$

With K$_w$, in mol kg$^{-1}$, the dissociation of water given by the adaptation of Millero's 1995 equation for freshwater:

$$lnKw = 148.9802 - \frac{13847.26}{T} - 23.6521\,lnT \quad\quad (6)$$

The first dissociation constant of carbonic acid is given by:

$K_1 = \dfrac{[H+][HCO_3^-]}{[H_2CO_3]}$ $\quad\quad\quad\quad$ (7)

and the second dissociation constant is given by:



$$K_2 = \frac{[H+][CO_3^{2-}]}{[HCO_3^-]} \qquad (8)$$

For low salinities, as in freshwater, the equations valid for low salinities (below a salinity of 5) from Millero (1995) can be used:

$$ln\,K_1 = 290.9097 - \frac{14554.21}{T} - 45.0575\,ln\,T + (-228.39774 + \frac{9714.36839}{T} + 34.485796\,ln\,T)S^{0.5} + (54.20871 -$$

$$\frac{2310.48919}{T} - 8.19515ln\,T)S + (-3.969101 + \frac{170.22169}{T} + 0.603627\,ln\,T)S^{1.5} - 0.00258768S^2 \qquad (9)$$

$$ln\,K_2 = 207.6548 - \frac{11843.79}{T} - 33.6485\,ln\,T + (-167.69908 + \frac{6551.35253}{T} + 25.928788\,ln\,T)S^{0.5} + (39.75854 -$$

$$\frac{1566.13883}{T} - 6.171951\,ln\,T)S + (-2.892532 + \frac{116.270079}{T} + 0.45788501\,ln\,T)S^{1.5} - 0.00613142S^2 \qquad (10)$$

With $K_1$ and $K_2$ in mol kg$^{-1}$ and T the temperature in Kelvin

The dissociation constants of TOC are given by Hruska et al., 2003:

$$K1_{TOC} = \frac{[H+][DOCH^{2-}]}{[DOCH_2^-]} = 10^{-4.51} \qquad (11)$$

$$K2_{TOC} = \frac{[H^+][DOC^{3-}]}{[DOCH^{2-}]} = 10^{-6.46} \qquad (12)$$

Solving analytically for protons the alkalinity equation (2) the polynomial resulting equation is:

$$[H^+]^6 + a[H^+]^5 + b[H^+]^4 + c[H^+]^3 + d[H^+]^2 + e[H^+] + f = 0 \qquad (13)$$

With,

$$a = Alk + K_1 + K1_{TOC} - TOC \qquad (14)$$

$$b = Alk * K1_{TOC} - K_w + K1_{TOC} * K2_{TOC} - 2 * K1_{TOC} * TOC + K_1 * K_2 + Alk * K_1 + K_1 * K1_{TOC} - K_1 * TOC - K_1 * DIC$$
(15)

$$c = Alk * K1_{TOC} * K2_{TOC} - K_w * K1_{TOC} - 3 * K1_{TOC} * K2_{TOC} * TOC + K_1 * K_2 * Alk + K_1 * K_2 * K1_{TOC} - K_1 * K_2 *$$
$$TOC + K_1 * Alk * K1_{TOC} - K_1 * K_w + K_1 * K1_{TOC} * K2_{TOC} - 2 * K_1 * K1_{TOC} * TOC - 2 * K_1 * K_2 * DIC + K1\_TOC * K_1 *$$
$$DIC \qquad (16)$$

$$d = K_1 * K_2 * Alk * K1_{TOC} - K_w * K1_{TOC} * K2_{TOC} - K_w * K_1 * K_2 + K_1 * K_2 * K1_{TOC} * K2_{TOC} - 2 * K_1 * K_2 * K1_{TOC} *$$
$$TOC + Alk * K1_{TOC} * K2_{TOC} * K_1 - K_w * K1_{TOC} * K_1 - 3 * K1_{TOC} * K2_{TOC} * K_1 * TOC - 2 * K_1 * K_2 * K1_{TOC} * DIC -$$
$$K_1 * K1_{TOC} * K2_{TOC} * DIC \qquad (17)$$

$$e = K_1 * K_2 * Alk * K1_{TOC} * K2_{TOC} - Kw * K1_{TOC} * K_1 * K_2 - 3 * K1_{TOC} * K2_{TOC} * K_1 * K_2 * TOC - Kw * K1_{TOC} *$$
$$K2_{TOC} * K_1 - 2 * K_1 * K_2 * K1_{TOC} * K2_{TOC} * DIC \qquad (18)$$

$$f = -Kw * K1_{TOC} * K2_{TOC} * K_1 * K_2 \qquad (19)$$

The Eq. (13) is solved using the Newton-Raphson algorithm.




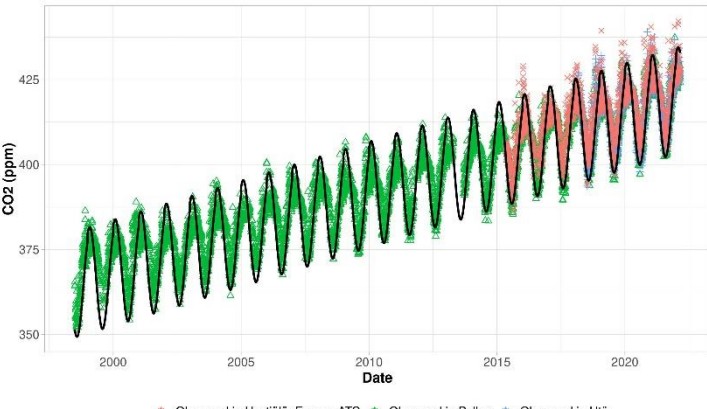

Observed in Hyytiälä_Europe ATC    Observed in Pallas    Observed in Utö

**Figure A2: Atmospheric CO₂ (ppm) as recorded at the Finnish weather stations of Hyytiälä (pink marker), Pallas (green marker) and Utö (blue marker) compared with the simulated atmospheric CO2 (black line, equation 6) (Source: Integrated Carbon Observation System ICOS RI, licensed under CC4BY)**


Table A1. Annual exports of TOC (kg ha$^{-1}$ a$^{-1}$) used in the first step of Vemala TOC model calibration

| Land use/soil class | kg ha$^{-1}$ a$^{-1}$ | Equation, comments | Reference |
|---|---|---|---|
| Peatland, with drainage | 159 | $-233.4+0.608*TS-0.0002349*TS^2$ | Finér et al. 2021 |
| Peatland, no drainage | 102 | $-117.8+0.359*TS-0.000145*TS^2$ | Finér et al. 2021 |
| Forest, mineral soils | 69 | $-122.5+0.277*TS-0.0001*TS^2$ | Finér et al. 2021 |
| Forest, mineral soils | 30-44 | some selected catchments | Mattsson et al. 2003 |
| Agriculture, mineral soils | 25-52 | | Manninen et al. 2018 |
| Agriculture, coarse soils | 32 | | Simulated from Löytaneenoja catchment (35_121) |
| Agriculture, peat soils | 161 | | Merja Myllys, personal communication |

* TS – Temperature sum (day degree, °C) = 1372 °C for Vantaanjoki catchment





Table A2: Land use and soil types in the upstream catchment of the tributaries and outlet of Tuusulanjärvi lake. With the soil types coloured blue for clay, grey for rock, orange for sand and green for till as characterized by the lower soil layer.

| Sub-catchment | Tributary | Agric. (%) | Urban (%) | Forest (%) | Water (%) | Predominant soil type 1 | Predominant soil type 2 | Predominant soil type 3 |
|---|---|---|---|---|---|---|---|---|
| 21.082 | Loutinoja | | 88 | 12 | | Vertic Cambisol (58%) | Dystric Leptosol (21%) | Haplic Podzol 2 (19%) |
| 21.082 | Räikilänoja | 38 | 56 | 6 | | Vertic Cambisol (75%) | Dystric Leptosol (23%) | Haplic Podzol 1 (1%) |
| 21.083 | Vuohikkanoja | 33 | 16 | 41 | 10 | Vertic Cambisol (40%) | Haplic Podzol 2 (29%) | Haplic Podzol 1 (5%) |
| 21.086 | Piiliojan va | 17 | 46 | 37 | | Vertic Cambisol (60%) | Dystric Leptosol (12%) | Haplic Podzol 2 (12%) |

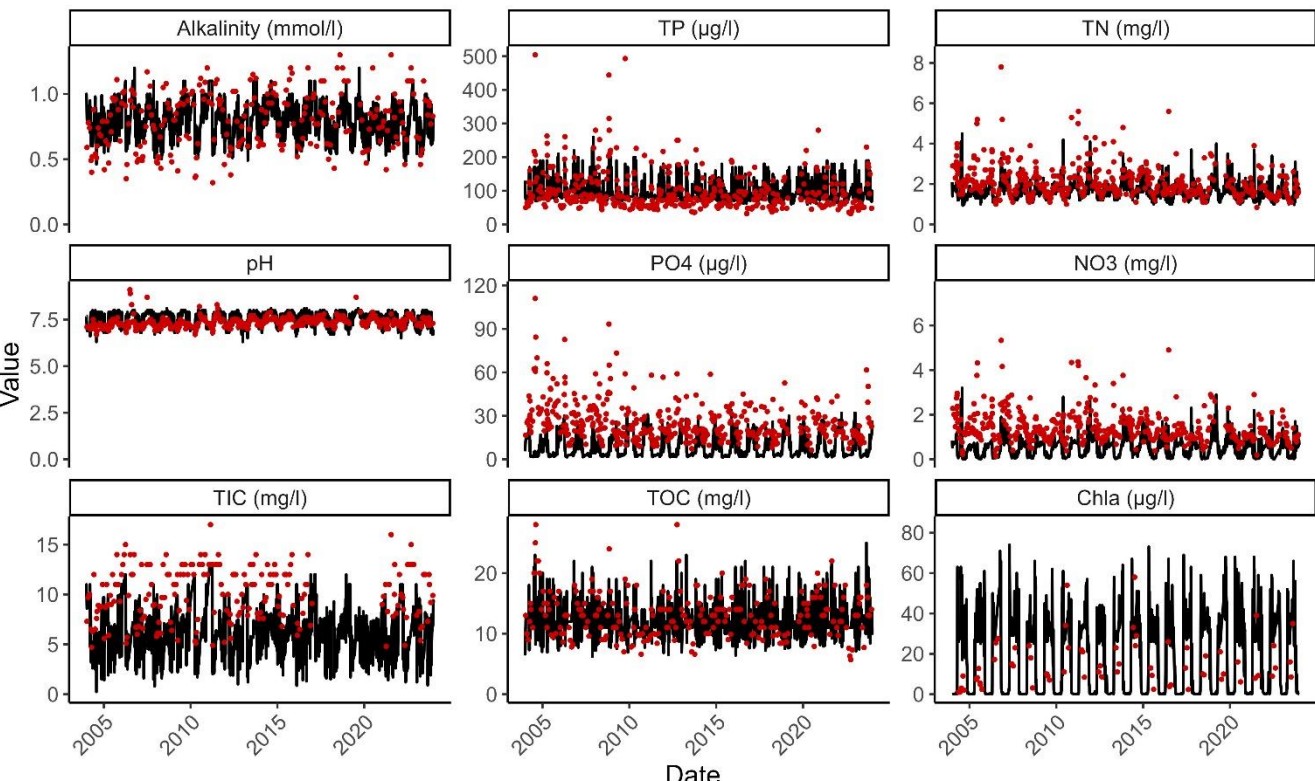

**Figure A3: Water quality in Vantaanjoki 4.2 6040 over the period 2004-2023 with observations (red dots) and**
**simulations (black line) for alkalinity (mmol L$^{-1}$), pH, total inorganic carbon (TIC, mg L$^{-1}$), total phosphorus (TP, µg L$^{-1}$), phosphate (PO$_4$, µg L$^{-1}$), total organic carbon (TOC, mg L$^{-1}$), total nitrogen (TN, mg L$^{-1}$), nitrate (NO$_3$, mg L$^{-1}$) and phytoplankton (Chla, µg L$^{-1}$).**