# Peer review of "Simulating carbon fluxes in boreal catchments: WSFS-Vemala model development and key insights"

_EGUsphere, 2025_

## Author Comment (AC2)

**Review of 'Simulating carbon fluxes in boreal catchments: WSFS-Vemala model development and key insights '**

Marie Korppoo[1], Inese Huttunen[1], Markus Huttunen[1], Maiju Narikka[1], Jari Silander[2], Tom Jilbert[3], Martin Forsius[4], Pirkko Kortelainen[4], Niina Kotamäki[5], Cintia Uvo[5,6], Anna-Kaisa Ronkanen[5]

Reviewer 2:

This work quantitatively predicts the dynamics of carbon flux at the scale of the river-lake aquatic ecosystem. The Vemala model was integrated into a hydrological model to simulate carbon flux driven by water flows. The evidence strongly supports their conclusions, and the findings hold great potential for providing new insights into the carbon cycle in river-lake systems.

*Answer: We thank reviewer 2 for the positive comment on the overall manuscript.*

The reviewers only raised minor concerns for the authors to consider in the revised version:
1. Details regarding observation data collection are insufficient, such as the methods for collecting water/soil samples and measuring target parameters.

*Answer: Thank you for this comment. We agree that the description of observational data should be clarified in the manuscript. The WSFS-Vemala model is developed and applied using long-term hydrological and water quality observations from national monitoring programmes maintained by the Finnish Environment Institute (Syke). Discharge data are obtained from the HYDRO dataset, which consists of continuous flow measurements from gauging stations across Finland. Water quality data, including total organic carbon, total inorganic carbon, alkalinity, pH, nitrogen, and phosphorus, are obtained from the VESLA dataset, which is based on routine grab sampling and laboratory analyses following standardized national protocols. Sampling frequency typically ranges from monthly to seasonal, depending on site and variable.*

*These observational datasets are used for model calibration and validation at the national scale. In addition, the targeted sampling campaign conducted in 2023 at Lake Tuusulanjärvi was used to support this study particularly. The sampling procedures and analytical methods for these measurements are described in detail in lines 290–292.*

*We will revise the Methods section accordingly to improve the clarity of the observational data description. As the current Methods section is already relatively long and detailed, we will also reconsider its overall structure and assess whether some technical details can be moved to Supplementary Materials. This will allow us to keep the main manuscript focused while still providing full methodological transparency. At the same time, we will ensure that all essential information regarding the observational datasets, sampling approaches, and analytical methods is clearly described in the main text.*

*HYDRO dataset / Syke, [https://metadata.ymparisto.fi/dataset/%7B86FC3188-6796-4C79-AC58-8DBC7B568827%7D](https://metadata.ymparisto.fi/dataset/%7B86FC3188-6796-4C79-AC58-8DBC7B568827%7D)*

*VESLA dataset / Syke, [https://ckan.ymparisto.fi/dataset/%7BB1444E19-0F36-49F5-A849-01A3D2083A11%7D](https://ckan.ymparisto.fi/dataset/%7BB1444E19-0F36-49F5-A849-01A3D2083A11%7D)*

2. Only a single reference is provided for Eqs. 13–14 in the text. Additional references should be added to solidify the selection of coefficients.

*Answer: Observations from the Vesla data management system of the Finnish Environment Institute (Syke) have been used. A reference to the database will be added to these equations.*

3. Time series plots comparing simulated and observed TOC/TIC are missing in Sections 4.1.2–4.1.3, which undermines the validity of the NSE values presented in Table 3.

*Answer: We agree that a reference to the appropriate figure already in the manuscript presenting TIC/TOC time series at the outlet of Vantaanjoki (suppl figure 3) is missing from this section 4.1.2. and 4.1.3. We will revise it accordingly.*

4. The limitations of the modeling approach require further discussion, such as the underlying assumptions of the dozens of sub-models.

*Answer: We agree that clarifying the assumptions and limitations is important. In the revised manuscript, we will discuss the need for further development of the model towards a combined inorganic/organic carbon soil model to take into account mineralisation in the soil as a source of inorganic carbon. Presently, the model simulates the inorganic carbon loading associated with rock weathering using alkalinity and soil types rather than accounting also for mineralisation of organic carbon in the soil. This model is seen as a first step towards a more integrated inorganic and organic carbon terrestrial model capable to simulate GHG from soils and TIC leaching from mineralisation and rock weathering. The lack of TIC data available for model development is the reason for the development of a separate TIC terrestrial sub-model. The strength of this sub-model is to use alkalinity data available at the national scale. Another limitation of the model is its applicability to areas dominated by carbonate soils. In boreal environments with rare carbonate soils, most of TIC and TOC are under a dissolved form. For the application of this model in carbonate soils, sedimentation processes of carbon should be added to the model.*
*The aim of this modelling approach is not a single lake but national scale modelling of TIC and TOC loading to the aquatic environment and to the Baltic Sea and to provide GHG from aquatic environments to be added to national estimates of GHG. We therefore focused our modelling development to the most important processes affecting TIC and TOC in boreal environments at the Finnish scale.*

**Citation**: https://doi.org/10.5194/egusphere-2025-3255-RC2

---

## Author Comment (AC3)

**Review of 'Simulating carbon fluxes in boreal catchments: WSFS-Vemala model development and key insights '**

Marie Korppoo[1], Inese Huttunen[1], Markus Huttunen[1], Maiju Narikka[1], Jari Silander[2], Tom Jilbert[3], Martin Forsius[4], Pirkko Kortelainen[4], Niina Kotamäki[5], Cintia Uvo[5,6], Anna-Kaisa Ronkanen[5]

**Reviewer 1:**

Overall, this is a well-written paper that describes an important modelling advance. While the model has been designed for Finnish conditions, the findings of this paper will be useful to a broad range of researches including those working with catchment or regional scale modelling, those interested in aquatic carbon cycling and climate issues as well as applied researchers having a responsibility to support decision makers. The authors present a regional / national scale model of aquatic carbon production, transport and loss. To the best of my knowledge, this is the first model to attempt such national scale simulations with such a high degree of process fidelity.
One of the key strengths of this model is that it tracks the production, transport, transformation and loss of both total inorganic carbon (TIC) and total organic carbon (TOC).

*Answer: We thank Reviewer 1 for the positive feedback on the manuscript and for acknowledging its clarity, novelty and relevance to the readers of Hydrology and Earth System Sciences.*

I do have a number of reservations about this paper that I hope the authors will have the opportunity to address in a revised version.
The authors present their model as a tool for simulating total organic carbon and total inorganic carbon. This is appropriate for boreal conditions where there is typically very little particulate organic carbon and the underlying geology for the most part precludes high levels of particulate inorganic carbon (e.g., carbonate –derived rocks).

*Answer: Thank you for highlighting the relevance of this model formulations under boreal conditions. In Finnish and other boreal catchments, particulate inorganic carbon is typically very low because carbonate rocks are rare and bedrock is dominated by silicate lithologies (Kortelainen et al., 2006* [https://link.springer.com/article/10.1007/s00027-006-0833-6](https://link.springer.com/article/10.1007/s00027-006-0833-6)*). Therefore, TIC can reasonably be assumed to represent dissolved inorganic carbon.*

*In contrast, the dominance of organic carbon in boreal inland waters is primarily controlled by land cover and climate rather than geology. Extensive peatlands and wetlands, organic-rich soils, and cold and humid climatic conditions promote the production, mobilisation, and export of dissolved organic carbon. As a result, DOC constitutes the dominant fraction of total organic carbon in boreal rivers and lakes. For example, Mattsson et al. (2005* [https://link.springer.com/article/10.1007/s10533-005-6897-x](https://link.springer.com/article/10.1007/s10533-005-6897-x)*) showed that, on average, 94% of TOC in Finnish rivers occurs in dissolved form.*

*We will explicitly add these details to the introduction in the revised version of the manuscript to clarify the applicability of this model under boreal conditions. We will also add in the discussion the limitation of the model applications to low particulate inorganic and organic carbon concentrations.*

I suggest the authors either refer to dissolved inorganic carbon (DIC) and dissolved organic carbon (DOC) throughout (as they seem to be doing from statements made on line 154), or note that in the environment for which this model has been developed, only a small fraction of the total aquatic carbon is in a particulate

form. Using DOC instead of TOC could also make more clear the separation between soil organic carbon and organic carbon in the aquatic phase.

*Answer: Thank you for this helpful suggestion. We agree that the use of DOC and DIC terminology can improve conceptual clarity, especially regarding the distinction between soil organic carbon and aquatic carbon pools. In our case, we use TOC and TIC because these are the forms in which long-term and spatially consistent observations are available for model development, calibration, and validation in Finland. We will add to the manuscript the link between non-carbonate bedrocks and low particulate inorganic carbon leading to the fact that most of the inorganic carbon in the water is under a dissolved form.*

*Ln 156: "TIC is assumed fully dissolved and thus is representing DIC."*

*Ln 289: "TOC and TIC are measured on unfiltered samples in Finland, while DOC and DIC are very rarely sampled."*

As the authors present their work as a new contribution to our ability to model aquatic carbon, I suggest deleting information about N and P simulations (e.g., Table 3). Either that or provide a rationale for why nitrogen and phosphorus simulation results should be included in this study.

*Answer: Thank you for this suggestion to focus the manuscript on carbon alone. The rationale for the presentation of nutrients results is linked to the application of the biogeochemical model concept in the aquatic ecosystem. The strength of the biogeochemical model is to simulate concurrently nutrients and carbon processes as they are combined through algal growth and mineralisation processes (sensitivity analysis 5.5 lines 569-575). We could add this rationale to the paragraph 2.3 Aquatic biogeochemical submodel. Presenting only TIC and TOC results would omit the strength of the model to simulate processes based on physical, chemical and biological reactions and how TIC and TOC are integrated with nutrients, and algal and bacterial growth. We believe this integration of nutrients and carbon is crucial in our work regarding the Water Framework Directive and the link between carbon and eutrophication.*

My biggest concerns about this paper arise from statements made on lines 134 and lines 153-157. On line 134, the authors state that "SOC and DIC can be mineralized into DIC that is simulated as a loss from the system to the air". Paraphrasing lines 153-157, they appear to state that alkalinity is a proxy for TIC which in turn includes $CO_2$, $HCO_3^-$ and $CO_3^{2-}$.

I would be grateful if the authors could clarify whether or not they are using the regression on line 156 to estimate the sum of $CO_2$, $HCO_3^-$ and $CO_3^{2-}$. If they are doing so, I would appreciate a stronger motivation for the decision.

*Answer: Thank you for this comment to clarify the model structure. The TOC terrestrial model is only used for the simulation of terrestrial loading of TOC and includes mineralisation in soils, however DIC storage in the soil is not explicitly simulated in this model version nor are $CO_2$ emissions from soil. The simulation of the TIC terrestrial loading that uses equation on ln156, and is related to alkalinity, is separate from the TOC loading model. This model simulates TIC as a bulk pool but does not simulate the carbonate speciation ($CO_2$, $HCO_3^-$, $CO_3^{2-}$) in the terrestrial ecosystem separately. The carbonate speciation is calculated only in the aquatic environment for the simulation of carbon emissions from the aquatic ecosystem and requires alkalinity, TIC and TOC for calculation of pH. To our understanding, geology through rock weathering is the main characteristic explaining the variation of TIC loading to surface waters. It is unclear to us what proportion of TIC leaching is explained by mineralisation in organic soils. A representation of national scale TIC and TOC concentrations in the aquatic environment (figure 1), shows that high TOC concentrations in Northern Finland (catchments number 36-67) are associated with low TIC concentrations. Northern catchments are till dominated catchments, which is a low source of alkalinity while southern catchments (5-30) are characterised by an increased proportion of clay and bedrock which are a higher source of alkalinity (Korkka-Niemi (2001)). Based on this assumption that rock weathering is the predominant process leaching TIC to surface waters, we built a model accordingly using geology and alkalinity as a proxy for TIC. Alkalinity simulations were also required in the aquatic ecosystem to calculate pH and simulate carbonate speciation and thus CO2 emissions.*

*Lines 135-137 should be modified to reflect the fact that DIC storage is not simulated explicitly in the TOC terrestrial model, only SOC and DOC dynamics (mineralisation, dissociation, storage and leaching) are simulated in the soil:*

'There are  two C storages in the soil – SOC _and_ DOC  linked to soil types and land uses. Inputs to the model are annual litter fall and initial C storage in soil. Interactions among these pools are as follows:

- SOC can be disassociated into DOC, and vice versa.

- SOC and DOC can be mineralized

- DOC leaches with the subsurface runoff and baseflow.'

[Figure]

*Figure 1: Average TIC and TOC concentrations in Finnish main catchments (VESLA database, Syke).*

My second concern about lines 134 and 153-157 is that they seem to state that terrestrial DIC is modelled twice, once as a breakdown product of SOC and / or TOC (line 134) and once as an empirical soil-related property (lines 153-157). Why? Doing so seems to violate a carbon mass balance as the DIC produced through mineralization leaves the system to contribute to atmospheric warming while the regression takes no explicit account of terrestrial carbon mineralization. This really needs a better explanation and justification.

*Answer: Thank you for this comment. There are historical reasons why TOC leaching is simulated separately from TIC leaching in Vemala. The Vemala model was originally developed to provide national-scale estimates of TOC loading and concentrations for the implementation of the Water Framework Directive (WFD). In Finnish waters, changes in TOC concentrations are known to contribute to brownification, which has ecological effects on inland water systems. When the model was first created (around 2018), TIC leaching was not considered a key output variable.*
*TIC leaching has since been integrated into Vemala as a separate sub-model for at least two main reasons:*
1. *The TOC module in Vemala does not simulate weathering processes in rocks and mineral soils. It only represents mineralization, which accounts for only part of the processes that generate TIC in soils.*
2. *The national scale observed TIC dataset is very small. However, the approach must rely on variables with national coverage for the development of the model at the national scale. Alkalinity is well correlated with TIC and is supported by comprehensive monitoring data across Finland.*
*In the future, the terrestrial TOC model could be coupled more tightly with the TIC model to simulate mineralisation processes and greenhouse gas emissions from land.*

From the text on lines 225-230, it appears that the authors calibrated to loads. This is poor practice for demonstrating the skill of a biogeochemical model. Any calibration that does a reasonable job of reproducing the observed flow has a high probability of generating misleadingly high Nash Sutcliffe

Efficiencies. Please consider either recalibrating to concentrations or present performance statistics based on modelled and observed concentations.

*Answer: Thank you for this important methodological comment. We agree that calibrating and evaluating biogeochemical models solely based on loads can be misleading, as loads are strongly controlled by discharge and may result in artificially high performance metrics such as NSE. In the WSFS-Vemala framework, calibration is not based on loads alone. The automatic calibration uses a modification of the direct search Hooke–Jeeves optimisation algorithm (Huttunen et al., 2016, https://link.springer.com/article/10.1007/s10666-015-9470-6) and considers both loads and concentrations during parameter optimisation.*

*To address this concern more explicitly, we will revise the manuscript to include performance statistics based on observed and modelled concentrations in the lake in addition to loads in the rivers. This will allow a more robust evaluation of model skill that is less dominated by discharge and better reflects biogeochemical process representation in lake systems. We will also clarify the distinction between calibration strategy and performance evaluation in the Methods and Results sections. Section 4.2.2 Water quality Tuusulanjärvi will be updated with the description of $r^2$ and PBIAS results in Tuusulanjärvi.*

Minor questions
L108 – how is soil temperature include in the model ? are measured time series used or is soil temperature simulated in some manner?

*Answer: Soil temperature is simulated within an unpublished soil frost simulation model developed in late 1990-ties in the WSFS system. The model is based on simulation of the energy flux between air, snow and soil layers. It calculates the snow thermal conductivity, soil thermal conductivity and soil specific heat. The model simulates distribution of the energy flux - how much energy is used:*
 *1) to freeze or melt the soil frost*
*2) to decrease or increase the soil temperature for the soil layers.*
*The Figure 2 below illustrates the soil temperature simulation for example at the 3rd level sub-catchment for years 2019-2020. The model is able to represent two quite different winter soil temperatures – snowy 2018/2019 and mild 2019/2020.*

[Figure]

*Figure 2. Soil temperature simulation for one 3rd level sub-catchment for years 2019-2020.*

L112-115 – Please provide some additional description of the Vemala conceptual model. After reading this text multiple times, it is still not entirely clear to me how the model represents the landscape. Is a watershed built up of "small brook catchments" or is some other approach used? I presume the model is semi-distributed as opposed to grid based? Having this type of background information would be quite useful to other modelers attempting to work at the same scale as Vemala.

*Answer: Thank you for this comment. An example of a map including 3$^{rd}$ level and small (4$^{th}$ level) brook sub-catchments can be added to supplementary materials for clarity. The model is semi-distributed and a better overall description of WSFS-Vemala system should be added below the 2 Model description section.*
*Over the whole Finland there are about 200 000 sub-catchments which are the simulation units of the model. These sub-catchments of Vemala represent an additional (4th) level of detail, created as a subdivision of the existing 3rd level sub-catchments dataset*
*([https://metadata.ymparisto.fi/dataset/{44394B13-85D7-4998-BD06-8ADC77C7455C}](https://metadata.ymparisto.fi/dataset/{44394B13-85D7-4998-BD06-8ADC77C7455C}) ). In Vantaanjoki, there are 48 3rd level sub-catchments split into 989 small (4th level) brook catchments of average size 148ha, excluding lake catchments.*
*Vemala simulates 40 different land-use/soil type combinations based on national soil and land cover datasets (Lilja et al. 2009 [https://urn.fi/URN:ISBN:978-952-487-252-2](https://urn.fi/URN:ISBN:978-952-487-252-2); Syke 2012 :*
*[https://metadata.ymparisto.fi/dataset/{66D9A881-EE3C-42AD-9416-014EA6B84D23}](https://metadata.ymparisto.fi/dataset/{66D9A881-EE3C-42AD-9416-014EA6B84D23}) ). Runoff is simulated separately for each combination at the 3$^{rd}$ level subcatchment scale (Kolhinen et al., 2026 [https://doi.org/10.1016/j.jhydrol.2025.134650](https://doi.org/10.1016/j.jhydrol.2025.134650)) and then used at the small brook subcatchment scale.*

L125 – Again, some more detail about the model structure would be appreciated. The authors note that carbon concentrations change with depth in both peat and mineral soils. Is this phenomenon represented in Vemala through different carbon contents in the unsaturated soil layer and groundwater layer?

*Answer: Carbon content is related to the depth according to the following equation adopted from Wen et al. 2019 (https://hess.copernicus.org/articles/24/945/2020/):*

$$C_d(z) = C_0 e^{\left(-\frac{z}{coef}\right)}$$

*where $C_d$ is SOC at z, the depth below the surface; $C_0$ is the SOC level at the*
*ground level and coef reflects the decline with depth, set here to a value of 0.3 for mineral soils. For peat soils coef=1.0 determining that there is no decline of SOC with depth. Thus, simulated SOC content is different in unsaturated soil layer and groundwater layer for mineral soils.*

L132- How are annual litter inputs added to the system? Are inputs prorated across every day of the year or is another approach used?

*Answer: The daily litter fall inputs are calculated from annual litter fall and are added to the soils during autumn months.*

L135 – I presume TOC produced in the leaching zone can percolate vertically to groundwater?

*Answer: Yes, it should be added to the manuscript that TOC produced in the unsaturated layer is percolated to the groundwater layer by percolated water, which is simulated in the hydrological model for each land use/soil texture class separately. Percolation of TOC is an important component of TOC balance in groundwater layer as it is one of the processes increasing TOC content in the groundwater layer, in contrast to the unsaturated soil layer, where litter fall is increasing OC content annually. However, simulation of percolation of TOC has caused also challenges in different soil textures, especially for coarse soils, where percolation is high due to the high hydraulic conductivity. In such soil DOC storage can be quickly emptied during the intensive snow melt or heavy autumn rainfall periods, when there are high amounts of percolation. In such cases the model simulated very low river TOC concentrations due to the probably*

*overestimated TOC percolation. Further discussion and developments are needed to better simulate TOC percolation in different soil textures in more realistic way.*

L145-148 – please provide numeric soil organic matter (SOM) levels for the SOM classification presented here; this information could be in the Supplementary Information

*Answer: The C content calculation is performed at the national scale since the WSFS-Vemala TOC model is applied at this scale. The C content calculation in mineral agricultural soils is based on field parcel data from Soil testing laboratory Viljavuuspalvelu oy which contains soil organic matter (SOM) class (vm - low, m – medium, rm –rich, erm – very rich, mm – mull, Tm – peat soil). Only the 5 first classes are for mineral soil and was used in creating C content for mineral soils. Only 40% of fields have observations, so the mean C content for 3rd level subcatchments for clay or for coarse soils was extrapolated based only on 40% of observed data.*

*Table 2. Soil organic matter classes according to soil type (mineral/organic) and soil organic matter percentage.*

| Orgaaninen aines, % *Organic matter, %* | Multavuusluokka *Organic matter class* | Lyhenne *Symbol* |
|---|---|---|
| < 3 | vähämultainen *low* | vm |
| 3–5,9 | multava *medium* | m |
| 6–11,9 | runsasmultainen *rich* | rm |
| 12–19,9 | erittäin runsasmultainen *very rich* | erm |
| 20–39,9 | multamaa *mull* | Mm |
| > 40 | turvemaa *peat soil* | Tm |

*The methodology was as follow:*
*1) the mean SOM content in % for the top soil for each class was obtained from LUKE report (Lemola et al., 2018, see the Table 2)*
*2) the area of agricultural clay soils and coarse soils for each 3rd level subcatchment was estimated,*
*3) the mean SOM content for clay and coarse soils separately was estimated, and then weighted mean SOM content for 3rd level subcatchments was estimated,*
*4) SOM content 1000 kg ha$^{-1}$ is calculated using bulk density of the mineral soils and SOM content,*
*5) it is assumed that SOM content in the 0-1 m deep soil (1000 kg ha$^{-1}$) is decreasing exponentially with the layer depth. Figure shows the national scale estimates of the OC content in the Finnish agricultural soils used as initial OC inputs to the Vemala TOC model. Corresponding values for Vantaanjoki catchment are 180-200 kg C/ha. This information can be added to the Supplementary material.*

[Figure]

Figure. Initial value of OC content in a) mineral agricultural soils (0-1 m) based on SOM class data (WSFS-Vemala)

L156 – how were the values in Table 2 obtained? Are they directly from Korka-Niemi (2001) or did the authors do the calibration themselves?

*Answer: The title of Table 2 should be updated stating the definition of the mean of the alkalinity per soil type as provided by Korkka-Niemi (2001) and rephrasing the sentence: "The range of alkalinities per soil types was defined from Korkka-Niemi (2001) measurements of well waters in Finland using the mean values per soil type with a range of ±20% (Table 2)." to "The range of alkalinities per soil types was defined using the mean values measured by Korkka-Niemi (2001) from well waters in Finland and a variation of ±20% from the mean values." The terrestrial loading of alkalinity was then calibrated within this range per soil type using the alkalinity observations available in the aquatic ecosystem.*

[Figure]

*Figure from Korkka-Niemi (2001).*

L165 – presumably a triprotic model is being used for DOC dissociation? Please identify which one. From statements made later in the manuscript, I presume it is the model of Hruska et al. (2003 https://pubmed.ncbi.nlm.nih.gov/12775041/)?

*Answer: The triptotic model used for the DOC dissociation refers to Hruska et al., 2003 work. A reference to Hruska et al., (2003) should be added to ln 165. Hruska's model is described in more details on ln 181 but a reference should be added earlier in the manuscript.*

L172 – phytoplankton settling in one of a number of processes that can lead to TOC sedimentation. Geochemical coagulation may be important in some circumstances. If phytoplankton settling in the only TOC process simulated in Vemala, please note that it may not be the only process operating in reality.

*Answer: Thank you for this comment. A sentence should be added in the manuscript that although phytoplankton settling is the only TOC process in the model feeding the sediments with TOC, it is not the only process occurring in the environment. The importance of iron in TOC sedimentation has been recognised in Finland (Heikkinen et al., 2022 https://doi.org/10.1016/j.scitotenv.2021.150256) and should be further studied before being added to the model at a later stage.*

Equations 3, 5 – please consider different left hand side terms for equations 3 and 5. It is a bit confusing to have them both described as "Alk" (I know there is the subscript "n" in equation 3 but that does not help terribly much)

*Answer: In equation 3 we defined the alkalinity load per soil type. The text describing the equation should be amended to add the term load to the alkalinity in ln 154 as well as the units used.  Equation 5 describes the total alkalinity in the water as a concentration, units in mmol L-1, should be added to the ln 189.*

Figures 3 b and c should be bigger if they are to be useful

*Answer: We agree with this statement. We tried to limit the number/size of the figures in the manuscript. We can provide larger maps for the final article or add more detailed maps including $3^{rd}$ level and $4^{th}$ level subcatchments in the supplementary materials for added clarity.*

Lines 295-300 – please provide more detail as to how flows at the Tuusulanjarvi outflow were estimated. Figure 4 – consider a separate plot for the Tuusulanjarvi outflow

*Answer: The outflow of Tuusulanjärvi is regulated and observed. We can modify Figure 4b to present observed outflow.*

Table 3 – please present NSE for concentrations, not loads in all cases.

*Answer: We can provide the statistical analysis for concentrations rather than for loads for all points.*

Line 485 – could the authors present any connection to PREBAS here? Is PREBAS simulating higher litter fall inputs over the study period and could tis account for the increase in DOC?  I would appreciate it if the authors could also comment on peat soil drainage as a factor behind increasing TOC. I was under the impression there was little or no new drainage of Finnish peat soils?

*Answer: Results of the forest growth model PREBAS were available for Vemala TOC modelling only for period 2017-2025, and for future scenarios. Therefore, literature values were used for the long-term tree biomass increase estimates for the Vemala TOC modelling. Main reference for that is Lehtonen and Heikkinen, 2016 (https://cdnsciencepub.com/doi/10.1139/cjfr-2015-0171). According to the Yasso07 model simulations there is about a 10% increase in total litter fall input to the soils. This information and reference can be added to the manuscript.*

**Fig. 2.** Some simulated time series (thin lines) and 95% confidence intervals computed from 500 simulated series for total litter input (top) and soil carbon stock changes (bottom) in (*a* and *c*) southern and (*b* and *d*) northern Finland.

[Figure]

*We agree with Reviewer 1 that there is practically no new peatland drainage performed in Finland. However, remedial drainage is still performed, and peatlands are managed for forest harvesting, which contribute to increased TOC leaching. We believe that already performed peatland drainage has possibly long-term effect on increasing TOC trends. Peatland drainage is causing higher concentrations in receiving streams compared to undrained peatlands, and possibly steeper increases over past decades from drained than undrained peatlands (Nieminen et al. 2021, https://www.sciencedirect.com/science/article/pii/S0048969721002163). This study is among the first which indicated that peatland drainage may have a long-term legacy effect on TOC concentrations. The results of this study also supported earlier findings in that the increase in forest cover and biomass ("greening effect") that has occurred in northern areas during the last decades may have contributed to increasing TOC trends.*

*Finer et al. 2021, (https://www.sciencedirect.com/science/article/pii/S0048969720376294) is writing – 'drainage for forestry has been shown to contribute to the increasing trends of OC and N fluxes in large river basins (Asmala et al., 2019; Räike et al., 2020). Drainage increases decomposition of surface peat and mineralization of organic matter as well as soil erosion, and therefore also the export of elements in both dissolved and particulate forms (Ahtiainen and Huttunen, 1999). These drainage impacts have been suggested to last – or even increase – over several decades after drainage (Nieminen et al., 2017, 2018).'*

*Some sentences summarizing the peatland drainage effect on TOC increasing trend and references can be added to the manuscript.*

Line 495 – What are the consequences, if any, of simulating alkalinity as a conservative tracer? It seems to imply that there will be no evasion of CO2 to the atmosphere but perhaps I misunderstand.

*Answer: Even though alkalinity is simulated as a tracer, the TIC concentrations are not and are affected by CO2 evasion to the atmosphere as well as mineralisation of TOC and primary production in the water column. Alkalinity is used to calibrate the terrestrial loading of TIC to the river/lake network and for the simulation of pH in relation with TIC and TOC. The definition of pH then leads to the calculation of the part of TIC that is dissolved as CO2 in the water and thus available for exchange with the atmosphere. Processes like photosynthesis, mineralisation, nitrification and denitrification in the water column affect alkalinity (e.g. Marescaux et al., 2020, https://doi.org/10.5194/hess-24-2379-2020) and thus would affect the pH simulations in the water. The importance of these processes on the overall alkalinity model would however be limited with findings from Marescaux et al. (2020) showing a contribution to alkalinity export from instream processes of less than 4%. At this stage of the model development, it is thus justified to simulate Alkalinity in the river network as a tracer.*

Figure A3 – In my opinion, Figure A3 is more convincing than Figure 7, why not switch these figures between the main text and SI?

*Answer: We would have liked to present both figures in the main text but for conciseness we placed the Figure A3 in the supplement. Figure 7 was needed for the validation of the lake processes and subsequent lake carbon budget discussed in the manuscript. Both figures could be kept in the main section of the manuscript.*

**Citation**: https://doi.org/10.5194/egusphere-2025-3255-RC1